# Heterophily-informed Message Passing

**Haishan Wang**                                             *haishan.wang@aalto.fi*
*Aalto University*

**Arno Solin**                                               *arno.solin@aalto.fi*
*Aalto University*

**Vikas Garg**                                               *vgarg@csail.mit.edu*
*YaiYai Ltd and Aalto University*

**Reviewed on OpenReview:** *https://openreview.net/forum?id=9fPinz1iH2*

## Abstract

Graph neural networks (GNNs) are known to be vulnerable to oversmoothing due to their implicit homophily assumption. We mitigate this problem with a novel scheme that regulates the aggregation of messages, modulating the type and extent of message passing locally thereby preserving both the low and high-frequency components of information. Our approach relies solely on learnt embeddings, obviating the need for auxiliary labels, thus extending the benefits of heterophily-aware embeddings to broader applications, e.g. generative modelling. Our experiments, conducted across various data sets and GNN architectures, demonstrate performance enhancements and reveal heterophily patterns across standard classification benchmarks. Furthermore, application to molecular generation showcases notable performance improvements on chemoinformatics benchmarks.

## 1 Introduction

Methods for ubiquitous graph-structured data with abundant topological information have advanced the field of graph representation learning in recent years. Graph neural networks (GNNs) have emerged as prominent deep learning models in this domain (Hamilton, 2020). A key feature of GNNs is the message-passing (MP) scheme—inspired by belief propagation—which facilitates the processing of local topology while maintaining computational efficiency (Dai et al., 2016). The MP scheme enables local information exchange between nodes and their neighbours; implicitly assuming strong *homophily*, *i.e.*, the tendency of nodes to connect with others that have similar labels or features. This assumption turns out to be reasonable in settings such as social data (McPherson et al., 2001), regional planning (Gerber et al., 2013), and citation networks (Ciotti et al., 2016). However, heterophilous graphs exist in many scenarios, *e.g.*, in fraud transaction networks (Pandit et al., 2007) and actor co-occurrence networks (Tang et al., 2009). They violate the homophily assumption, leading to sub-optimal performance (Zhu et al., 2020; 2021; Chien et al., 2021; Lim et al., 2021; Wang et al., 2023), owing to *oversmoothing* (Li et al., 2018) resulting from flattening of high-frequency information (Wu et al., 2023) by MP schemes.

A conceptual way to characterize homophily is by examining the neighbours of each node in a graph. For example, in a graph representing a molecule a fully *homophilous* molecule only has links between atoms of the same type, while a *heterophilous* molecule has links between different types. However, in practice, node labels necessary for homophily calculation are often missing due to lack of information or unavailable due to privacy concerns. Instead, heterophily typically stems from more intricate properties of the graph which need to be learned from data. For this problem, we introduce a general heterophily-informed MP scheme to carefully address and utilize the heterophily properties in graph data.

Many previous works analyze the impact of heterophily on GNN performance and design innovative structures to mitigate it (Zhu et al., 2020; Liu et al., 2021a; Yan et al., 2022; Ma et al., 2021). Some studies offer deeper insights into how heterophily affects model expressiveness (Ma et al., 2021; Luan et al., 2022; Mao et al.,

2023; Luan et al., 2023). However, most of these works provide a specialized model and only focus on simple node classification tasks. In contrast, we aim to design a simple yet flexible structure that can be applied to any GNN. We verify its effectiveness not only on classification tasks but also on more challenging molecular generation tasks, which require models to learn the data distribution on graph embeddings.

**Our contributions**  We introduce a novel heterophily-informed MP scheme, serving as an approach for general GNNs to utilize data heterophily, designed to learn graph structures and node features across varying degrees of homophily and heterophily. The scheme improves various GNN architectures for node classification in different domains of data. Furthermore, we apply our approach to a flow-based graph generation model (HetFlows). Our key contributions are summarized below:

- **Conceptual and technical:** We propose an architecture-independent approach that encodes homophily/heterophily patterns for general GNNs with flexible applications and highlights the necessity of data heterophily as the model prior.

- **Methodological:** We design a heterophily-informed message-passing scheme focusing on specific node heterophily patterns and apply it to *(i)* various classic GNNs for node classification and *(ii)* an invertible flow-based model (HetFlows) for molecule generation.

- **Empirical:** We demonstrate the benefits of our idea by benchmarking node classification accuracy and in molecule generation by evaluating the generated molecules with an extensive set of chemoinformatics metrics.

Notable advantages of our model include enhancing performance on different graph learning tasks without adding extra parameters, offering flexible applications across various GNN architectures, and revealing the homophily pattern match between message passing and data sets.

## 2 Related Work

**Graph heterophily**  Graph heterophily refers to the connectivity tendency among nodes with different labellings, as opposed to graph homophily. This property could be measured by node homophily (Pei et al., 2019), edge homophily (Zhu et al., 2020), or more dedicated designed metrics (Lim et al., 2021; Platonov et al., 2023). Recent research shows the importance and benefits of considering heterophily in graph learning. For example, Empirical experiments (Zhu et al., 2022) show heterophilic nature in structure attacks, and validate that heterophily incorporation enhances model robustness to adversarial attacks. Similarly to oversmoothing, the heterophily challenges graph learning by less discriminative node representation, Bodnar et al. (2022) links the GNN performances in heterophilic graphs with the oversmoothing problem by cellular sheaf theory.

**Heterophily-aware GNNs**  Numerous techniques have been developed to address degradation in the performance of GNNs in heterophilic settings. Some approaches expand the concept of neighbour sets, such as aggregating messages from farther hops of neighbours (Abu-El-Haija et al., 2019; Wang & Derr, 2021) or searching for potential new neighbours (Pei et al., 2019; Jin et al., 2021; Li et al., 2022). Other focus on refining the message during the aggregation process, such as differentiating neighbours through specific filters (Luan et al., 2022; Yan et al., 2022; Wang et al., 2023) or collecting embeddings from previous layers, like jumping knowledge (Xu et al., 2018) and generalized PageRank techniques (Chien et al., 2021). These methods typically require specialized structures or ignore the local homophily during message passing to achieve their effects. However, we hope to capture the local homophily difference with minimal structure modifications, making the solution flexible and widely applicable. Heterophily-informed MP can be viewed as performing adaptive message modulation before aggregation.

**Molecule representation and generation**  Early works in molecule generation (Kusner et al., 2017; Guimaraes et al., 2017; Gómez-Bombarelli et al., 2018; Dai et al., 2018) primarily used sequence models to encode the SMILES (short for 'Simplified Molecular-Input Line-Entry System') strings (Weininger et al., 1989), posing generation as an autoregressive problem. However, the mapping from molecules to SMILES is

not continuous, so similar molecules can be assigned vastly different string representations. Graphs provide an elegant abstraction to encode the interactions between the atoms in a molecule. Thus the field has gravitated towards representing molecules as (geometric) graphs and using powerful graph encoders; *e.g.*, based on graph neural networks (GNNs), for example, adversarial models (De Cao & Kipf, 2018; You et al., 2018), energy-based models (Liu et al., 2021b), diffusion models (Hoogeboom et al., 2022; Xu et al., 2023), Neural ODEs (Verma et al., 2022) and flow-based models (Shi et al., 2019; Luo et al., 2021; Zang & Wang, 2020).

## 3 Heterophily-informed Message Passing

We propose an architecture-independent approach that encodes homophily/heterophily patterns for general GNNs and later apply it to GNNs in node classification and graph generation setups.

### 3.1 Prerequisites: Message Passing

**Graph Neural Networks** (GNNs) have emerged as a potent paradigm for learning from graph-structured data, where the challenges include diverse graph sizes and varying structures (Kipf & Welling, 2017; Veličković et al., 2018; Xu et al., 2019; Garg et al., 2020). Consider a graph $G = (\mathcal{V}, \mathcal{E})$ with nodes $\mathcal{V}$ and edges $\mathcal{E}$. For these nodes and edges, we denote the corresponding node features as $\boldsymbol{X} = \{\boldsymbol{x}_v \in \mathbb{R}^{n_a} \mid v \in \mathcal{V}\}$ and edge features as $\boldsymbol{E} = \{\boldsymbol{e}_{uv} \in \mathbb{R}^{n_b} \mid u, v \in \mathcal{E}\}$. Here $n_a, n_b$ are the feature dimensions of nodes and edges. For each node $v \in \mathcal{V}$, its embedding at the $k^{\text{th}}$ layer is represented as $\boldsymbol{h}_v^{(k)}$, and $\boldsymbol{h}^{(k)} = \{\boldsymbol{h}_v^{(k)} \mid v \in \mathcal{V}\}$. These embeddings evolve through a sequence of transformations across GNNs of depth $K$, by the message passing scheme (Hamilton, 2020):

$$\boldsymbol{m}_{uv}^{(k)} = \text{MESSAGE}^{(k)}\left(\boldsymbol{h}_u^{(k)}, \boldsymbol{e}_{uv}\right), \quad u \in \mathcal{N}(v), \tag{1}$$

$$\boldsymbol{h}_v^{(k+1)} = \text{UPDATE}^{(k)}\left(\boldsymbol{h}_v^{(k)}, \boldsymbol{m}_{\mathcal{N}(v)}^{(k)}\right), \tag{2}$$

for $k \in \{0, 1, \ldots, K\}$. Here $\mathcal{N}(v)$ denotes the neighbour set of node $v$. Both $\text{UPDATE}^{(k)}$ and $\text{MESSAGE}^{(k)}$ are arbitrary differentiable functions. Messages from all neighbours of $v$ are aggregated in the multiset $\boldsymbol{m}_{\mathcal{N}(v)}^{(k)} = \{\{\boldsymbol{m}_{uv}^{(k)} \mid u \in \mathcal{N}(v)\}\}$. Importantly, the function $\text{UPDATE}^{(k)}$ needs to be permutation invariant on this message set $\boldsymbol{m}_{\mathcal{N}(v)}^{(k)}$ (*e.g.*, by resorting to operations like summation or taking the maximum). However, in practice, a naïve aggregation strategy typically mixes messages, especially in heterophilic locality, leading to the 'oversmoothing' problem (Zhu et al., 2020; 2021; Chien et al., 2021; Lim et al., 2021; Wang et al., 2023).

### 3.2 Heterophily-informed Message Passing

Our method encodes the heterophily assumption into the MP scheme of the GNN, denoted as $\text{GNN}^\gamma(\boldsymbol{X} \mid \boldsymbol{E}) = \boldsymbol{h}^{(K)}$. The indicator $\gamma \in \Gamma = \{\text{orig.}, \text{hom.}, \text{het.}\}$ specifies the heterophily preference of the GNNs as a hyperparameter: whether they lean towards homophily (hom.), heterophily (het.), and original structure (orig.). The $\text{GNN}^{\text{orig.}}$ is exactly the original GNN described at Sec. 3.1. And we name $\text{GNN}^{\text{het.}}$ and $\text{GNN}^{\text{hom.}}$ after the HetMP and HomMP modes of $\text{GNN}^{\text{orig.}}$. Referring to Eq. (1) and Eq. (2), the messages undergo different scaling preprocessing steps before being sent forward to the subsequent layer:

$$\boldsymbol{m}_{uv}^{(k)} = \text{MESSAGE}^{(k)}\left(\boldsymbol{h}_u^{(k)}, \boldsymbol{e}_{uv}\right), \quad u \in \mathcal{N}(v), \tag{3}$$

$$\boldsymbol{m}_{\mathcal{N}(v)}^{(k)} = \{\{\alpha_{uv,\gamma}^{(k)} \boldsymbol{m}_{uv}^{(k)} \mid u \in \mathcal{N}(v)\}\}, \tag{4}$$

$$\boldsymbol{h}_v^{(k+1)} = \text{UPDATE}^{(k)}\left(\boldsymbol{h}_v^{(k)}, \boldsymbol{m}_{\mathcal{N}(v)}^{(k)}\right), \tag{5}$$

where the scaling factors

$$\alpha_{uv,\gamma}^{(k)} = \begin{cases} \mathcal{H}(u, v), & \text{if } \gamma = \text{hom.} \\ 1, & \text{if } \gamma = \text{orig.} \\ 1 - \mathcal{H}(u, v), & \text{if } \gamma = \text{het.} \end{cases} \tag{6}$$

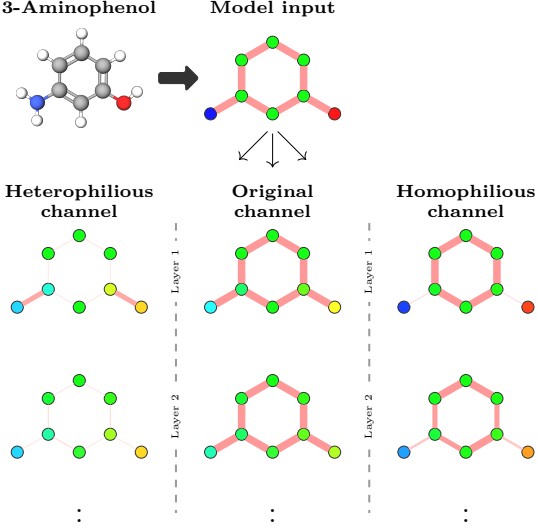

Figure 1: Comparison of heterophiliy-informed MP with original GNN on a graph describing the 3-Aminophenol molecule. The three channels show how $\gamma$ controls the scaling factor in Eq. (6) and leads to different message passing behaviour, given the same input.

and $\mathcal{H}$ denotes the node homophily and the embeddings are initilized by $\boldsymbol{h}^{(0)} = \boldsymbol{X}$. Aiming to learn embeddings as node labels, therefore in practice, instead of traditional label-style definition in many contexts, we define the homophily or *attraction to similarity* between embeddings as the cosine similarity $\mathcal{H}(u, v) \triangleq S_{\cos}(\boldsymbol{h}_u^{(k)}, \boldsymbol{h}_v^{(k)})$ at the relevant layer in Eq. (6). For more representative embeddings, three channels (HomMP, HetMP, and original model) could be combined with an additional linear layer to be a mixed model (mix.) named MixMP. App. A.5 provides an example of GCN-based MixMP.

The message passing process of an example GNN and its HetMP, HomMP versions are visualized in Fig. 1 as three channels. The input is an example molecule 3-Aminophenol, and node colouring corresponds to the embeddings while similar colour means closer embeddings. The thickness of the bond corresponds to the scaling factor $\alpha_\gamma$ (which shows the similarity between node pairs) of the last layer. The homophilous channels (hom.) decrease information exchange between dissimilar neighbours, introducing frictions during message flows. The heterophilous channel (het.) encourages faster message spreading on higher frequency locality, while slowing it down when neighbors become similar. In this simple example, the original channel (orig.) shows the typical oversmoothing issue: all the node embeddings become similar after layers, while the other two channels (homophilous/heterophilous channels) mitigate it with different convergence resistances, thus improving model effectiveness.

### 3.3 Application to Molecular Generation

For generative modelling on molecule graphs, we propose a graph-generation model based on heterophily-informed MP: HetFlows. The model is built on a normalizing flow-based model MoFlow (Zang & Wang, 2020), which generates molecules with estimated probability and ensures uncertainty estimation and applications of Bayesian tools. Our model is split into two main components: a bond flow and an atom flow. The bond flow learns the molecular topology, while the atom flow assigns certain atomic details to this structure.

**Prerequisites** In general, **normalizing flows** offers a methodological approach to model distributions based on the change-of-variable law of probabilities. This is achieved by applying a chain of reversible and bijective transformations between trivial variables (like Gaussians) with target data variables and updating the transformation to minimize the negative log-likelihood (Dinh et al., 2014).

**Affine coupling layers** (ACLs) introduce reversible transformations to normalizing flows, ensuring efficient computation of the log-determinant of the Jacobian (Kingma & Dhariwal, 2018). ALCs keep reversibility

by updating partial information via the other part of the information. Series of ACLs $\{f_1, \ldots, f_T\}$ build a reversible flow between the Gaussian with the target distribution $f = f_T \circ \cdots \circ f_1$.

**Training and loss**   Given molecule graph $G = (\boldsymbol{X}, \boldsymbol{E})$, the atom flow $f_a$ and bond flow $f_b$ map the graph into embeddings which follow Gaussian distributions $\mathcal{N}_a, \mathcal{N}_b$:

$$f_a(\boldsymbol{X} \mid \boldsymbol{E}) = \boldsymbol{h}_a \sim \mathcal{N}_a, \quad f_b(\boldsymbol{E}) = \boldsymbol{h}_b \sim \mathcal{N}_b. \tag{7}$$

The model is trained to minimize the negative log-likelihoods (NLLs) of the data by gradient descent. The target loss function of the model $\mathcal{L}$ could be decomposed into two parts since HetFlows contains two flows $\mathcal{L} = \mathcal{L}_a + \mathcal{L}_b$. The atom loss $\mathcal{L}_a$ is defined as following

$$\mathcal{L}_a = -\log p_X(\boldsymbol{X}) = -\log p(\boldsymbol{h}_a) - \log \det \left( \left| \frac{\partial f_a}{\partial \boldsymbol{X}} \right| \right). \tag{8}$$

The bond loss $\mathcal{L}_b$ is defined similarly as above.

**Generation process**   Given a trained model $f_{a*}, f_{b*}$ with established parameters, sampled embeddings randomly from Gaussian $\boldsymbol{h}_a, \boldsymbol{h}_b \sim \mathcal{N}_a, \mathcal{N}_b$. Then the embeddings can generate features of bonds $\boldsymbol{E} = f_{b*}^{-1}(\boldsymbol{h}_b)$ and atoms $\boldsymbol{X} = f_{a*}^{-1}(\boldsymbol{h}_a \mid \boldsymbol{E})$ in sequence, which requires the reversibility of flow model. Finally, the molecules are reconstructed $G = (\boldsymbol{X}, \boldsymbol{E})$. Additionally, the bond features generation can be achieved by sampling the adjacency matrix (denoted by '+true adj.' later) from the real data distribution which reflects the model separability.

The HetFlows shares the same structure of MoFlow (Zang & Wang, 2020): ACL-based normalizing flows. The coupling function stores most of the parameters of an ACL, which serves as the most essential part. The main difference between these two methods is the message passing scheme of GNN utilized as coupling functions in all ACLs of flow: MoFlow contains the classical graph convolutional networks (Kipf & Welling, 2017), and the HetFlows substitute it with a MixMP version of it. Further details of HetFlows are discussed in App. A, including introductions of normalizing flows and ACLs, mathematical model description, the training and generation process of HetFlows, the loss function, and proof of model reversibility.

# 4   Experiments

We demonstrate the effects of heterophily-informed MP both in discriminative node classification benchmarks and molecule generation settings. Sec. 4.1 compares the node classification performances of various types of GNNs with their variants (HetMP, HomMP, and MixMP versions) across 15 data sets. In Sec. 4.2, we demonstrate how our MixMP GCN blocks improve flow-based molecule generation, directly showing the benefits of our MP scheme.

## 4.1   Node Classification Benchmarks

**Data sets**   The node classification experiments belong to the class of semi-supervised learning tasks. We evaluated on 5 homophilic data sets in citation networks (Yang et al., 2016) (CORA, PUBMED, CITESEER) and co-purchase graphs (Shchur et al., 2018) (COMPUTERS, PHOTO). Furthermore, the 10 heterophilic data sets including hyperlink networks (Pei et al., 2019) (CORNELL, WISCONSIN, TEXAS), Wikipedia networks (Rozemberczki et al., 2021) (CHAMELEON, SQUIRREL), and heterophilous graph dataset (Platonov et al., 2023) (ROMAN-EMPIRE, AMAZON-RATINGS, MINESWEEPER, TOLOKERS, QUESTIONS). Here the graph homophily $\mathcal{H}_{ei}$ is measured by class insensitive edge homophily ratio (Lim et al., 2021).

**Setups**   The models were implemented in PyTorch (Paszke et al., 2019) and PyTorch Geometric (PyG) (Fey & Lenssen, 2019). The base model GNNs in Sec. 4.1 are the PyG-implemented versions. Each one and its variants (HetMP, HomMP) contain 2 layers and 128 dimensions for all hidden layers. All the models are trained with the AdamW optimizer (Loshchilov & Hutter, 2019), learning rate 0.001 and drop-out ratio 0.2.

Table 1: We shed light on how our homo-/heterophilous message passing can boost standard node classification benchmark accuracy (mean±std). Data sets are sorted according to their homophily level $\mathcal{H}_{ei}$. For each data (column), four classic GNN architectures (GCN, GAT, GIN, GraphSAGE) are tested, grouped with their corresponding HetMP/HomMP/MixMP modes (with +hom., +hom. or +mix.). Numbers are bolded with a paired $t$-test (5% significance level) for each 4-mode group. Our MixMP models perform either equally well or significantly better in all cases, with an average of 3.84 (%-points) accuracy improvement than the original model. Note that the accuracy of binary class datasets is reported as their ROC-AUC scores, which follows the convention from Platonov et al. (2023).

| | TEXAS | MINESWEEPER | ROMAN-EMPIRE | SQUIRREL | CORNELL | CHAMELEON | QUESTIONS | WISCONSIN | AMAZON-RATINGS | TOLOKERS | CITESEER | PUBMED | COMPUTERS | CORA | PHOTO |
|---|---|---|---|---|---|---|---|---|---|---|---|---|---|---|---|
| Homophily $\mathcal{H}_{ei}$ | 0.00 | 0.01 | 0.02 | 0.03 | 0.06 | 0.06 | 0.08 | 0.10 | 0.13 | 0.18 | 0.63 | 0.66 | 0.70 | 0.77 | 0.77 |
| GCN | **58.4**±4.6 | 52.3±0.6 | 47.8±0.8 | 26.9±1.4 | 42.7±4.9 | 37.1±3.0 | **53.2**±0.4 | 52.4±6.0 | **49.1**±0.6 | 58.0±2.9 | **76.7**±1.3 | 88.5±0.5 | **91.6**±0.5 | **88.0**±0.9 | 94.4±0.5 |
| GCN+hom. | **58.6**±5.1 | 57.3±1.0 | 56.6±0.6 | 29.6±1.7 | 41.4±5.4 | **40.4**±3.0 | 52.8±0.2 | **54.5**±3.3 | 47.1±0.5 | **57.7**±3.2 | **76.8**±1.4 | **88.7**±0.5 | **91.5**±0.5 | 86.9±0.9 | **94.4**±0.4 |
| GCN+het. | **60.0**±7.7 | 71.5±1.8 | 60.2±0.5 | 28.8±1.3 | **50.0**±8.2 | 28.6±1.5 | 52.1±0.9 | **53.9**±3.8 | 46.4±0.6 | 55.1±1.1 | 69.6±1.2 | 83.9±0.5 | 78.3±1.4 | 82.6±1.1 | 85.4±1.6 |
| GCN+mix. | 56.8±5.3 | **74.5**±2.3 | **71.5**±0.5 | **32.0**±2.7 | **48.9**±6.2 | 38.9±3.9 | **53.1**±0.6 | 52.0±6.1 | **49.7**±0.8 | **58.7**±1.6 | **76.9**±1.1 | **88.6**±0.5 | 91.2±0.6 | **87.7**±0.7 | **94.6**±0.6 |
| GAT | **57.8**±4.6 | 54.4±1.6 | 47.1±0.6 | 29.4±1.5 | 45.7±6.3 | 43.3±2.9 | 52.4±0.4 | 50.8±8.7 | 48.8±0.5 | 57.0±2.7 | **76.2**±1.1 | 87.1±0.5 | **91.2**±0.3 | **86.6**±0.9 | **94.1**±0.5 |
| GAT+hom. | **59.5**±4.9 | 55.5±3.2 | 57.6±1.0 | 32.6±1.5 | 47.0±7.5 | 43.6±2.4 | 53.0±0.7 | **54.9**±5.8 | 47.2±0.7 | 55.7±3.2 | 75.9±1.4 | **88.2**±0.5 | 90.9±0.3 | 85.9±1.1 | 93.6±0.6 |
| GAT+het. | **59.7**±5.5 | 52.7±3.8 | 48.0±1.3 | 29.4±1.1 | **58.9**±9.4 | 27.9±1.6 | 51.3±0.8 | **53.3**±6.8 | 44.1±0.6 | 50.5±0.5 | 67.5±1.7 | 79.6±0.9 | 84.0±2.0 | 75.1±1.4 | 87.3±1.3 |
| GAT+mix. | **59.2**±4.7 | **75.1**±2.4 | **68.9**±0.9 | **34.4**±1.4 | 50.5±7.5 | **46.4**±2.6 | **54.8**±0.9 | **55.3**±6.3 | **50.2**±0.5 | **62.4**±2.7 | **75.7**±1.3 | 87.8±0.7 | **91.1**±0.3 | 85.7±1.4 | **94.1**±0.3 |
| GraphSAGE | **73.2**±5.3 | **73.2**±1.7 | **78.9**±0.4 | **36.5**±1.9 | **70.5**±3.7 | 50.2±1.9 | 56.0±0.8 | **77.1**±6.5 | 52.7±0.6 | **60.5**±1.6 | 76.7±1.0 | 88.1±0.5 | **90.9**±0.3 | **87.7**±1.5 | **95.5**±0.4 |
| GraphSAGE+hom. | 70.5±5.5 | 68.4±1.4 | **78.6**±0.6 | **36.2**±1.6 | **67.3**±4.1 | **50.8**±2.6 | 55.7±0.7 | **80.2**±5.2 | 51.6±0.5 | **59.7**±1.5 | **77.4**±0.7 | **88.9**±0.4 | 90.6±0.4 | **87.5**±1.1 | **95.5**±0.4 |
| GraphSAGE+het. | **75.9**±6.8 | 71.0±1.9 | 78.1±0.7 | 35.5±1.4 | **72.2**±7.6 | **50.7**±2.4 | 56.1±0.7 | **80.0**±4.3 | 52.2±0.6 | 57.7±1.4 | 76.0±0.8 | 87.7±0.5 | 89.4±0.6 | 86.6±1.3 | 95.0±0.6 |
| GraphSAGE+mix. | **73.0**±6.5 | **73.1**±1.9 | **78.7**±0.7 | **36.4**±2.3 | **68.1**±6.5 | **50.6**±2.5 | **57.2**±1.2 | **80.2**±5.0 | **53.5**±0.7 | 59.2±2.0 | 76.5±0.9 | 88.4±0.5 | 90.3±0.5 | **87.6**±0.8 | 95.2±0.5 |
| GIN | **57.0**±6.9 | 55.1±3.2 | 48.2±1.2 | 26.5±3.1 | **46.8**±7.1 | **36.8**±5.6 | 51.8±3.0 | 44.9±6.8 | **50.2**±0.7 | **53.3**±6.9 | **71.0**±1.5 | 86.1±0.8 | 41.2±3.1 | **84.0**±1.1 | 35.3±7.1 |
| GIN+hom. | **58.1**±6.8 | 58.2±4.6 | **65.9**±0.8 | 24.6±1.0 | **46.8**±6.6 | 36.0±3.8 | **55.7**±4.6 | **49.0**±6.3 | 49.2±0.7 | **53.0**±6.4 | **71.2**±1.9 | **87.4**±0.8 | 42.8±3.8 | **84.3**±1.3 | 37.7±8.0 |
| GIN+het. | **59.5**±8.0 | **65.7**±9.8 | 58.6±1.1 | 26.5±2.0 | **50.3**±10.7 | 35.9±5.6 | 50.8±2.6 | **52.0**±3.8 | 48.2±1.2 | **53.7**±5.8 | 69.3±1.7 | **81.1**±14.9 | **43.1**±9.3 | 81.9±1.1 | 38.4±15.7 |
| GIN+mix. | **58.4**±6.8 | **60.5**±7.0 | 64.0±1.0 | **28.9**±1.8 | 49.5±6.7 | 36.5±4.1 | **59.3**±3.9 | 50.6±7.8 | 48.2±1.5 | **56.3**±7.0 | **72.6**±2.2 | 86.6±0.9 | **57.3**±14.7 | **84.3**±1.5 | **78.2**±13.6 |

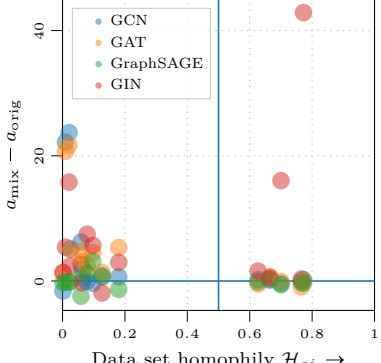

(a) Accuracy improvement of MixMP

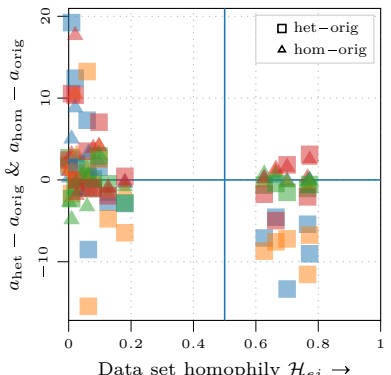

(b) Accuracy improvement of HetMP and HomMP

Figure 2: Comparison over 15 data sets and 4 GNN architectures in terms of improvement in accuracy (%-points). $a_{\mathrm{orig}}$, $a_{\mathrm{hom}}$, $a_{\mathrm{het}}$ and $a_{\mathrm{mix}}$ denote the accuracy of original model and corresponding HomMP, HetMP, and MixMP versions. In Fig. 2a, each node is the average performance over 10 random seeds for a specific dataset and GNN structure, it illustrates the accuracy advantage of MixMP above the original model. In Fig. 2b, the $y$-axis denotes the accuracy improvement of HetMP and HomMP over the original structure.

The data split settings (training/validation/test = 60%/20%/20%). Each configuration (data set and model) is tested for 10 random model initializations and data splits.

**Baselines** Four different GNN architectures are chosen as baseline models: Graph Convolutional Network (GCN) (Kipf & Welling, 2017), Graph Attention Network (GAT) (Veličković et al., 2018), Graph Isomorphism Network (GIN) (Xu et al., 2019), and GraphSAGE (Hamilton et al., 2017). For more details about the data split rules and base GNNs, please refer to App. B.

**Results** The comprehensive benchmark results in Table 1 include the node classification accuracy (or ROC-AUC score for binary labels), each value representing the average of 10 random runs of the four base models and their HetMP, HomMP, and MixMP versions across 15 different graph domains, with data homophily displayed. For each GNN and data set, the best result among the four modes is highlighted in bold. For more intuitive visualizations, Fig. 2 illustrates the benefits of classification accuracy on heterophily-informed MP and special patterns through the accuracy discrepancies. Further comparison with other heterophily-aware baseline models is in App. B.4.

**Analysis**   Table 1 demonstrates that, across 10 heterophilic data sets, our proposed MixMP performs comparably or better than the baseline in 38 out of 40 cases, evaluated over four types of GNN architectures. This strong performance is further supported by Fig. 2a, where the majority of data points lie on or above the $y$-axis, indicating that the MixMP consistently enhances node classification performance. Notably, the improvement is stronger on heterophilous datasets compared to homophilous ones (with minor outliers from GIN), suggesting that our model modification is able to incorporate data heterophily as structural prior without benefits from homophilic data. The consistent performance gains across diverse setting combinations underscore the potential generalization capability of our approach, which could also be applied to a wider range of GNN structures and data domains.

In Fig. 2b, the accuracy differences $a_{het} - a_{ori}$ and $a_{hom} - a_{ori}$ are mostly concentrated in the top-left and bottom-right regions. For homophilic data, the HomMP exhibits a minor impact on accuracy, whereas the HetMP even adversely affects performance. In contrast, heterophilic data benefit from both the HomMP and HetMP on the node classification task. These observations provide empirical evidence for the implicit homophily assumption underlying the traditional MP scheme. Furthermore, both the HomMP and HetMP mitigate the issue of oversmoothing in heterophilic data, as visualized in Fig. 1.

Minor performance improvements are observed on the Cornell, Wisconsin, and Texas datasets, likely attributed to their small graph size (183–251 nodes) with limited statistical significance. In contrast, our MixMP achieves substantial significant improvements on the Minesweeper and Roman-empir, which are specifically designed to evaluate GNNs under heterophily (Platonov et al., 2023). These results further validate the advantages of our approach in scenarios where heterophily is a critical factor.

In conclusion, the node classification experiments highlight the capability of MixMP to incorporate data heterophily as prior and enhance node-level graph representation learning, especially on heterophilic data.

## 4.2   Molecule Generation

Molecules express varying levels of homo-/heterophily. Thus molecular modelling provides an interesting benchmark for our proposed MP scheme. We demonstrate the impact of accounting for heterophilic message passing in a variety of common benchmark tasks for molecule generation and modelling. We provide results for molecule generation with benchmarks on a wide range of chemoinformatics metrics.

**Setups**   The HetFlows in Sec. 4.2 is built on GNNs with 4 layers and flows that were $k_a = 27, k_b = 10$ (for QM9) deep and $k_a = 38, k_b = 10$ (for ZINC-250K). For the generation task, we select the best-performing model using the FCD score as suggested in Polykovskiy et al. (2020). We report numbers without *post hoc* validity corrections.

**Data sets**   We consider two common molecule data sets: QM9 and ZINC-250K. The QM9 data set (Ramakrishnan et al., 2014) comprises ∼134k stable small organic molecules composed of atoms from the set {C, H, O, N, F}. These molecules have been processed into their kekulized forms with hydrogens removed using the RDkit software (Landrum et al., 2013). The ZINC-250K (Irwin et al., 2012) data contains ∼250k drug-like molecules, each with up to 38 atoms of 9 different types.

**Chemoinformatics metrics**   We compare methods through an extensive set of chemoinformatics metrics that perform both sanity checks (validity, uniqueness, and novelty) on the generated molecule corpus and quantify properties of the molecules: neighbour (SNN), fragment (Frag), and scaffold (Scaf) similarity, internal diversity (IntDiv$_1$ and IntDiv$_2$), and Fréchet ChemNet distance (FCD). We also show score histograms and distribution distances for solubility (logP), synthetic accessibility (SA), drug-likeness (QED), and molecular weight. For computing the metrics, we use the MOSES benchmarking platform (Polykovskiy et al., 2020) and the RDKit open-source cheminformatics software (Landrum et al., 2013). The 'data' row in metrics is based on averages over 10 randomly sampled sets (1000 mols per set) from the data. For the metrics, we simulate 10 batches of 1000 mols and compare them to a hold-out reference set (20% of data, other 80% used for training). Full details on the 14 metrics we use are included in App. D.

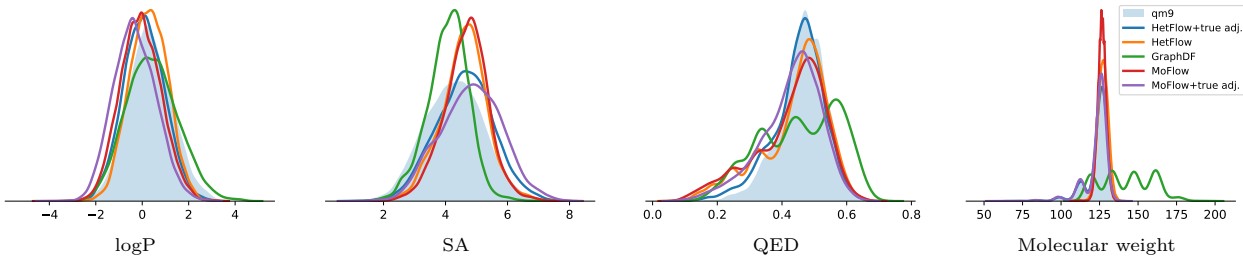

Figure 3: Chemoinformatics statistics for data (QM9) and generated molecules from HetFlows (ours), MoFlow, and GraphDF. We report histograms for the Octanol-water partition coefficient (logP), synthetic accessibility score (SA), quantitative estimation of drug-likeness (QED), and molecular weight.

Table 2: Chemoinformatics summary statistics for random generation on the QM9 molecule data. Full listing of all 14 metrics in Table A6. The results show that our message passing modification of MoFlow (resulting in HetFlows) achieves better results on FCD, SNN, Frag, and Scaf, and retains competitive performance on other metrics.

| | | FCD ↓ | Validity ↑ | Novelty ↑ | SNN ↑ | Frag ↑ | Scaf ↑ | IntDiv$_1$ ↑ |
|---|---|---|---|---|---|---|---|---|
| | Data (QM9) | 0.40 | 1.00±0.00 | 0.62±0.02 | 0.54±0.00 | 0.94±0.01 | 0.76±0.03 | 0.92±0.00 |
| Flows | GraphDF | 10.76 | - | **0.98**±0.00 | 0.35±0.00 | 0.61±0.01 | 0.09±0.07 | 0.87±0.00 |
| | MoFlow | 7.55 | 0.95±0.01 | 0.96±0.01 | 0.32±0.00 | 0.60±0.03 | 0.04±0.03 | **0.92**±0.00 |
| | HetFlow (Ours) | 4.04 | 0.92±0.01 | 0.92±0.01 | 0.34±0.00 | 0.80±0.02 | 0.04±0.03 | 0.91±0.00 |
| Abl. | MoFlow+true adj. | 4.45 | **1.00**±0.00 | 0.85±0.01 | 0.38±0.00 | 0.70±0.03 | 0.31±0.08 | **0.92**±0.00 |
| | HetFlow+true adj. | **1.46** | **1.00**±0.00 | 0.74±0.01 | **0.43**±0.00 | **0.85**±0.02 | **0.52**±0.05 | **0.92**±0.00 |

Table 3: Chemoinformatics summary statistics for random generation on the ZINC-250K molecule data set. Full listing of all 14 metrics in Table A7.

| | | FCD ↓ | Validity ↑ | Novelty ↑ | SNN ↑ | Frag ↑ | Scaf ↑ | IntDiv$_1$ ↑ |
|---|---|---|---|---|---|---|---|---|
| | Data (ZINC-250K) | 1.44 | 1.00±0.00 | 0.02±0.00 | 0.51±0.00 | 1.00±0.00 | 0.28±0.02 | 0.87±0.00 |
| Flows | GraphDF | 34.30 | - | **1.00**±0.00 | 0.23±0.00 | 0.35±0.01 | 0.00±0.00 | **0.88**±0.00 |
| | MoFlow | 23.33 | 0.89±0.01 | **1.00**±0.00 | 0.27±0.00 | 0.79±0.00 | 0.01±0.00 | **0.88**±0.00 |
| | HetFlow (Ours) | 23.72 | 0.87±0.01 | **1.00**±0.00 | 0.26±0.00 | 0.77±0.01 | 0.01±0.00 | **0.88**±0.00 |
| Abl. | MoFlow+true adj. | **8.21** | **0.94**±0.01 | **1.00**±0.00 | 0.33±0.00 | 0.89±0.00 | 0.07±0.02 | 0.87±0.00 |
| | HetFlow+true adj. | 8.24 | 0.93±0.01 | **1.00**±0.00 | **0.34**±0.00 | **0.91**±0.01 | **0.10**±0.03 | 0.87±0.00 |

**Baselines** For random generation, we include baseline results for methods that have pre-trained models publicly available. Trained models are required for generating chemoinformatics metrics beyond trivial sanity checks (validity, uniqueness, and novelty). We compare GraphDF (Luo et al., 2021) and MoFlow (Zang & Wang, 2020) which are current state-of-the-art flow models for molecular generation.

**Results on** QM9 For the QM9 data set, the main chemoinformatic summary statistics are given in Table 2 and the descriptive distributions in Fig. 3. HetFlows +true adj. achieves best validity, SNN Frag and Scaf, especially lowest FCD over flow-based models. Full benchmark results are available in the extended listings in Table A6 in the Appendix.

**Results on** ZINC-250K For the ZINC-250K data set, the main chemoinformatic summary statistics are given in Table 3 and the descriptive distributions in Fig. A7. While the ZINC-250K data set is more complicated, HetFlows +true adj. achieves the best Novelty SNN, Frag and Scaf, and has competitive performance on other metrics. Full benchmark results are available in the extended listings in Table A7.

**Analysis** HetFlows emerges as a robust and versatile molecular generation model, adept at balancing fidelity, diversity, and molecular properties. Notably, the validity on both QM9 and ZINC-250K are higher than 85%, ensuring the model's reliability. As discussed in Sec. 3.3, HetFlows is built based on the MoFlow (Zang &

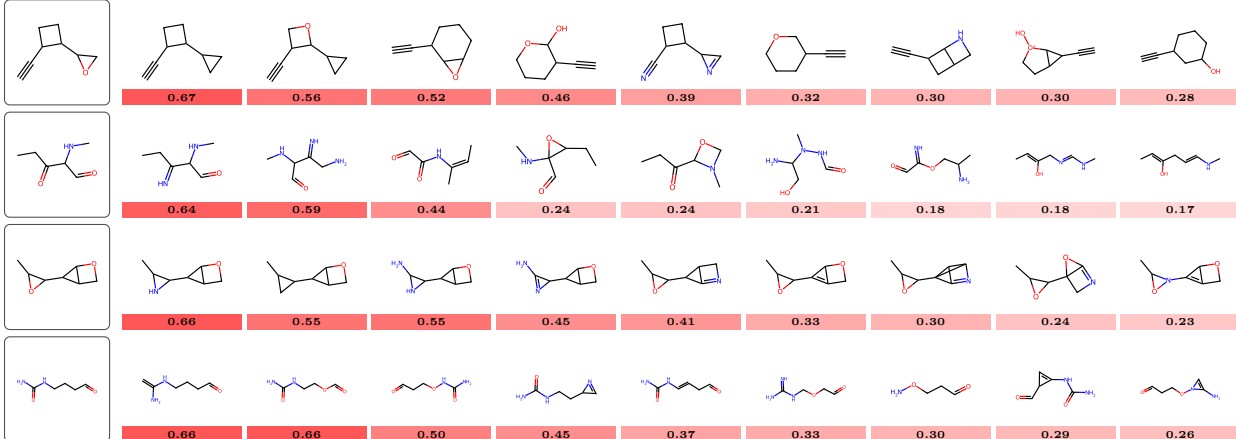

Figure 4: **Structured latent-space exploration (**QM9**).** Our approach yields a sensibly structured latent space as qualitatively demonstrated by this nearest neighbour search in the latent space with the seed molecule on the left and neighbours with the Tanimoto similarity (1 █████ 0) given for each molecule. For results on ZINC-250K, see Fig. A6 in the appendix.

Wang, 2020). It replaces the classic graph convolution of the GCNs module of MoFlow with MixMP version. In other words, the difference between MoFlow and HetFlows is the GNNs, more exactly, the MP scheme. As Table 2 and Table 3 show, HetFlows is superior over MoFlow with both adjacency matrix generation styles on QM9 and ZINC-250K. Fig. 3 shows the chemoinformatics statistics of generated molecules from HetFlows fit the original distribution better than MoFlow in both generation styles. It provides evidence of the advantage of utilizing the heterophily inside the message passing, as intuited by Fig. 1.

**Visualizing the continuous latent space**  Inspired by Zang & Wang (2020), we examine the learned latent space of our method on both QM9 and ZINC-250K, presented in Fig. 4 and Fig. A6, respectively. Qualitatively, we note that the latent space appears smooth and the molecules near the seed molecule resemble the input and have high Tanimoto similarity (Rogers & Hahn, 2010).

## 4.3   Ablations Studies

To further verify that the effects we see are due to our proposed heterophily-informed MP scheme, we include ablation studies on different adjacency matrix generation strategies and parameter sharing of the GNNs.

**Ablation study: adjacency matrix generation**  As mentioned in Sec. 3.3, the adjacency matrix can be generated by the bond model or sampled directly from the real distribution. Both approaches are present in literature as a basis for modelling (see, *e.g.*, discussion in Verma et al., 2022). With sampled adjacency matrices, the main task of the model is to put correct labels on a given graph topology. As the comparison results in Table 2 and Table 3 show, the adjacency matrix generated by the bond model limits the model generation performance compared with the sampling alternative. However, this approach can be considered a method of its own.

**Ablation study: parameter sharing of MixMP GNN**  There are three channels of message passing scheme as shown in Fig. 1, the three channels could share parameters or not in the MixMP, which corresponds to whether $\text{MESSAGE}_\gamma^{(k)}$ in Eq. (3) is the same function for all $\gamma \in \Gamma$ or not. To investigate the improved expressiveness of MixMP GNNs from extra two channels, models with two settings are compared in Table A8 and Table A9. Based on the chemoinformatics metrics, the random generation outputs of the two settings are similar. It means the model strength stems from the more expressive structure as intuited by Fig. 1, not from the larger parameter size.

## 5 Conclusions and Discussion

We have presented a heterophily-informed message-passing scheme, a flexible plug-in module of GNNs to account for a heterophily prior, countering the traditional oversmoothing vulnerability prevalent in existing GNN-based methodologies. By adjusting message passing to discern (dis-)similarities between nodes, our method offers a more nuanced representation of the intricate balance between affinities and repulsions. In the experiments, we demonstrated our approach both in standard discriminative node classification benchmarks and by applying the approach inside a generative flow model (which we call HetFlows). Experiment results show the versatility and ability of the proposed scheme to enhance embedding expressiveness across multiple graph domains. The analysis underscores the necessity of aligning data homophily with corresponding model assumptions. We consider this approach a promising tool in heterophilic graph learning.

One limitation of heterophily-informed MP is the variable effectiveness across datasets of different homophily levels. As discussed in Sec. 4.1, the scheme brings limited advantages to all homophily data and heterophily data of small sizes. It also suggests clear potential for improving various MP mechanisms. Such improvements could be achieved through better estimation of graph homophily and more intelligent integration of different channels.

In molecular generation, the dependency on the current adjacency matrix generation (bond model) method perhaps restricts HetFlows's effects, as the key coupling functions, based on CNNs, capture limited structural information. With a sampled adjacency matrix, it successfully generates molecules that are valid, novel, diverse, and chemoinformatically similar to those in the existing distribution.

A reference implementation of the methods is available at `https://github.com/AaltoML/heterophily-imp`.

### Acknowledgments

This work was supported by the Research Council of Finland (grants 342077, 362408, 339730), Saab-WASP (grant 411025), and the Jane and Aatos Erkko Foundation (grant 7001703). We acknowledge the computational resources provided by the Aalto Science-IT project and CSC – IT Center for Science, Finland.

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

## Appendices

This appendix is organized as follows. App. A presents more about HetFlows: Prerequisite knowledge of normalizing flows and affine coupling layers, training process, generation process, loss function, details of MixMP GNNs in HetFlows, computational issues and reversibility proof of the model. App. B illustrates related details of node classification experiment, including the experiment environment, data sets information and message passing details of GNNs used as base models. App. C provides the experiment environment, data sets, additional experiment results for molecule generation, property optimization algorithm with corresponding results together and model selection details. App. D summarizes and describes the metrics used in the molecular generation experiments.

## A    HetFlows Details

This section presents all the technical details of HetFlows.

### A.1    Prerequisites: Normalizing Flows with Affine Coupling Layers

**Affine coupling layers** (ACLs) introduce reversible transformations to normalizing flows, ensuring efficient computation of the log-determinant of the Jacobian (Kingma & Dhariwal, 2018). Typically, the affine coupling layer, denoted by $\text{ACL}^{(f,\boldsymbol{M})}$, contains a binary masking matrix $\boldsymbol{M} \in \{0,1\}^{m \times n}$ and a **coupling function** $f$ which determines the affine transformation parameters. The input $\boldsymbol{X} \in \mathbb{R}^{m \times n}$ is split into $\boldsymbol{X}_1 = \boldsymbol{M} \odot \boldsymbol{X}$ and $\boldsymbol{X}_2 = (\boldsymbol{1} - \boldsymbol{M}) \odot \boldsymbol{X}$ by masking, where '$\odot$' denotes the Hadamard (element-wise) product. Here, $\boldsymbol{X}_1$ is the masked input that will undergo the transformation, and $\boldsymbol{X}_2$ is the part that provides parameters for this transformation via the coupling function and stays invariant inside the ACLs. The output is the concatenation of the transformed part and the fixed part, as visualized in Fig. A5 given as:

$$\text{ACL}^{(f,\boldsymbol{M})}(\boldsymbol{X}) = \boldsymbol{M} \odot (\boldsymbol{S} \odot \boldsymbol{X}_1 + \boldsymbol{T}) + (\boldsymbol{1} - \boldsymbol{M}) \odot \boldsymbol{X}_2, \tag{9}$$

such that $\log \boldsymbol{S}, \boldsymbol{T} = f(\boldsymbol{X}_2)$. The binary masking ensures that only part of the input is transformed, allowing the model to update certain features while fixing others, enabling the model's reversibility. The coupling functions of flow capture intricate data characteristics, into which we incorporate heterophily priors.

**Normalizing flows** offers a methodological approach to model distribution based on the change-of-variable law of probabilities. This is achieved by applying a chain of reversible and bijective transformations between trivial variables (like Gaussian) with target data variables and updating the transformation to minimize the negative log-likelihood (Dinh et al., 2014). Given a target distribution $\boldsymbol{z}_0 \sim p_{\boldsymbol{z}}$, we initialize flows $f = f_T \circ \cdots \circ f_1$. The flow reach trivial varibales $\boldsymbol{z}_T \sim \text{N}(\mu, \sigma^2)$ through a series of invertible functions: $\boldsymbol{z}_i = f_i(\boldsymbol{z}_{i-1}), i = 1, 2, \ldots, T$. The goal of normalizing flows is to minimize the negative log-likelihoods (NLLs) of the data:

$$\mathcal{L} = -\log p_{\boldsymbol{z}}(\boldsymbol{z}_0) = -\log \text{N}(\boldsymbol{z}_T \mid \mu, \sigma^2) - \log \det \left| \frac{\partial f}{\partial \boldsymbol{z}_0} \right|. \tag{10}$$

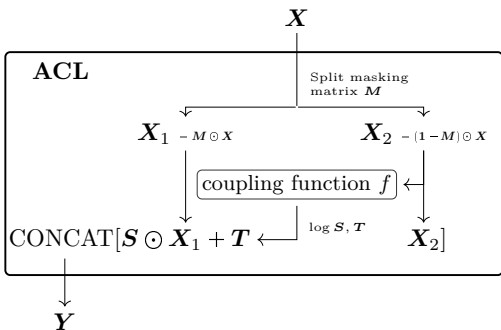

Figure A5: The affine coupling layer. The coupling is defined through a coupling function $f$ and binary masking matrix $\boldsymbol{M}$ (seen in Eq. (9)).

The power of normalizing flows lies in their bijectiveness. This ensures that no information from the data is lost during these transformations. Thus, the transformed distribution can be 'pulled back' to the original space using the inverse of the transformation functions, providing a bridge between the Gaussian and the intricate target distribution.

## A.2 Training Process

Molecule graph $G$ contains atom (node) features $\boldsymbol{X} \in \mathbb{R}^{n \times n_a}$ and bond (edge) features $\boldsymbol{E} \in \mathbb{R}^{n \times n \times n_b}$. The terms $n_a$ and $n_b$ denote the types amount of atoms and bonds respectively. And $(\boldsymbol{X})_i$ denotes the one-hot encoded type of the $i^{\text{th}}$ atom present in molecule $G$. Similarly, $(\boldsymbol{E})_{ij}$ denotes the one-hot encoding of the specific chemical bond between the $i^{\text{th}}$ and $j^{\text{th}}$ atoms. Our model HetFlows maps the molecule $G$ to embeddings $\boldsymbol{h}_a$ and $\boldsymbol{h}_b$ that follow the Gaussian distributions:

$$\boldsymbol{h}_a \sim p_a = \mathrm{N}(\mu_a, \sigma_a^2), \quad \boldsymbol{h}_b \sim p_b = \mathrm{N}(\mu_b, \sigma_b^2). \tag{11}$$

**Bond flow**    The bond flow represented by $f_b = \mathrm{ACL}_{k_b}^b \circ \cdots \circ \mathrm{ACL}_1^b$ consists of a series of ACLs with convolutional neural networks (CNNs) as coupling function: $\mathrm{ACL}_i^b = \mathrm{ACL}^{(\mathrm{CNN}_i, \boldsymbol{M}_i^b)}, \boldsymbol{M}_i^b \in \mathbb{R}^{n \times n \times n_b} \quad i = 1, 2, \ldots, k_b$, where $k_b$ denotes the number of layers. Then bond embeddings $\boldsymbol{h}_b = \boldsymbol{h}_b^{(k_b)} = f_b(\boldsymbol{h}_b^{(0)})$ are updated by layers:

$$\boldsymbol{h}_b^{(i)} = \mathrm{ACL}_i^b \left( \boldsymbol{h}_b^{(i-1)} \right), \quad i = 1, 2, \ldots, k_b. \tag{12}$$

initialized from the bond tensor $\boldsymbol{h}_b^{(0)} = \boldsymbol{E}$

**Heterophilous atom flow**    The atom flow $f_a$ contains $k_a$ affine coupling layers. Each layer consists of a masking matrix $\boldsymbol{M}$ and a GNN coupling function $\mathrm{GNN}^\Gamma$.

$$\mathrm{ACL}_i^a = \mathrm{ACL}^{(\mathrm{GNN}_i^\Gamma, \boldsymbol{M}_i)}, \quad i = 1, 2, \ldots, k_a. \tag{13}$$

where $(\boldsymbol{M}_i)_{j,k} = \mathbf{1}_{j \equiv i(n)}$, and $\Gamma = \{\text{orig., hom., het.}\}$ indicate MP scheme as mentioned in Sec. 3.1. $\mathrm{GNN}^\Gamma$ is a MixMP GNN combining these three schemes as described at App. A.5. All GNNs in this context derive their graph topology $(\mathcal{E}, \boldsymbol{E})$ from the bond tensor $\boldsymbol{E}$. The embeddings are initialized by the atom features: $\boldsymbol{h}_0^{(a)} = \boldsymbol{X}$, and undergo an update through each affine coupling layer as follows:

$$\boldsymbol{h}_a^{(i)} = \mathrm{ACL}_i^a \left( \boldsymbol{h}_a^{(i-1)} \mid \boldsymbol{E} \right), \quad i = 1, 2, \ldots, k_a. \tag{14}$$

The final node embedding is $\boldsymbol{h}_a = \boldsymbol{h}_a^{(k_a)} = f_a(\boldsymbol{h}_a^{(0)})$.

**Training target**    The training goal of HetFlows is to decrese the distance between the transformed variables $\{\boldsymbol{h}_a, \boldsymbol{h}_b\}$ with target distribution $p_a, p_b$, as assumed in Eq. (11). The loss function combines the NLLs from both the atom and bond flow: $\mathcal{L} = \mathcal{L}_a + \mathcal{L}_b$ (details at App. A.4). Each NLL is given as shown in Eq. (10). The target of training is to search for model parameters to minimize the loss:

$$f_{a*}, f_{b*} = \arg\min_{f_a, f_b} \mathcal{L} \tag{15}$$

## A.3 Generation Process

Given a trained HetFlows model, with established atom flow $f_{a*}$ and bond flow $f_{b*}$, the procedure for generating molecules is described as follows ($*$ denotes parameters fixed).

1. **Sampling Embeddings:** Start by randomly sampling embeddings $\boldsymbol{h}_a \sim p_a$ and $\boldsymbol{h}_b \sim p_b$ from a Gaussian distribution as expressed in Eq. (11).

2. **Obtaining the Bond Tensor:** The bond tensor $\boldsymbol{E}$ can be derived by applying the inverse of the bond flow $f_{b*}^{-1}$ to the sampled embedding $\boldsymbol{h}_b$. This is given as $\boldsymbol{E} = f_{b*}^{-1}(\boldsymbol{h}_b)$.

3. **Recovering Graph Topology:** From the bond tensor $\boldsymbol{E}$, the graph topology $(\mathcal{E}, \boldsymbol{E})$ can be deduced. This topology is essential for the node feature generation at the next step.

4. **Generating Node Features:** With the bond tensor in place, the generated node features can be produced by applying the inverse of the atom flow $f_{a*}^{-1}$ to the sampled atom embedding $\boldsymbol{h}_a$ given by $\boldsymbol{X} = f_{a*}^{-1}(\boldsymbol{h}_a \mid \boldsymbol{E})$.

5. **Molecule Recovery:** Finally, a molecule, represented as $G = (\boldsymbol{X}, \boldsymbol{E})$, can be reconstructed from random embeddings $[\boldsymbol{h}_a, \boldsymbol{h}_b]$.

The topology generation process (Steps 2–3) can be achieved by sampling the adjacency matrix from the real (data) distribution, which is denoted by '+true adj.'

**Reversibility of HetFlows**

To ensure that the molecular embeddings produced by HetFlows can be inverted back, it is crucial to understand the reversibility of the processes. A formal proof of reversibility of ACL blocks and HetFlows is provided in App. A.7.

## A.4 Loss function

The loss function of the HetFlows is the NLLs of flows $\mathcal{L} = \mathcal{L}_a + \mathcal{L}_b$ consists of atom flows and bond flows. The details are listed as follows: The bond model loss $\mathcal{L}_a$ comes from the NLLs as the Eq. (10)

$$\mathcal{L}_b = -\log p\left(\boldsymbol{h}_b\right) - \sum_{i=1}^{k_b} \log \det\left(\left|\frac{\partial \mathrm{ACL}_i^b}{\partial \boldsymbol{h}_b^{(i-1)}}\right|\right). \tag{16}$$

Similarly, the loss $\mathcal{L}_a$ for the atom flow can be constructed as:

$$\mathcal{L}_a = -\log p\left(\boldsymbol{h}_a\right) - \sum_{i=1}^{k_a} \log \det\left(\left|\frac{\partial \mathrm{ACL}_i^a}{\partial \boldsymbol{h}_a^{(i-1)}}\right|\right). \tag{17}$$

## A.5 The MixMP Graph Neural Network

In the code implementation, the MixMP GCN, $\mathrm{GNN}^\Gamma$ ($\Gamma = \{\mathrm{orig.}, \mathrm{hom.}, \mathrm{het.}\}$), consists of convolutional layers, batch normalization layers and a series of linear layers. The convolutional layer is the improved version of the graph convolutional layer of GCNs (Kipf & Welling, 2017), with three channels of MP. Each channel transfers messages between neighbours by certain preferences based on the homophily/heterophily.

Given input $\boldsymbol{h} \in \mathbb{R}^{n \times n_a}$ and bond tensor $\boldsymbol{E} \in \mathbb{R}^{n \times n \times n_b}$, assume the adjacency matrix $\boldsymbol{A}_i = \boldsymbol{E}[:,:,i] \in \mathbb{R}^{n \times n}$, $i = 1, \dots, n_b$. Here $n_a, n_b$ are the feature dimensions of the node and bond. Then the messages are scaling with the scaling matrix $\boldsymbol{H}_\gamma \in \mathbb{R}^{n \times n}$, and $(\boldsymbol{H}_\gamma)_{ij} = \alpha_{ij,\gamma}$. The homophily factors $\alpha_{uv,\gamma}$ differ from channels indicated by $\gamma \in \Gamma$

$$\alpha_{uv,\gamma} = \begin{cases} \mathcal{H}(u,v), & \text{if } \gamma = \mathrm{hom.} \\ 1, & \text{if } \gamma = \mathrm{orig.} \\ 1 - \mathcal{H}(u,v), & \text{if } \gamma = \mathrm{het.}, \end{cases} \tag{18}$$

and $\mathcal{H}(u,v) \triangleq S_{\cos}(\boldsymbol{h}_u^{(k)}, \boldsymbol{h}_v^{(k)})$ is the cosine similarity. The node embeddings are updated by the transferred messages in these triple channels, for each edge channel $i$,

$$\hat{\boldsymbol{h}}_i = \mathrm{cat}\left(\boldsymbol{H}_{\mathrm{orig.}} \odot \boldsymbol{A}_i \boldsymbol{h}, \boldsymbol{H}_{\mathrm{hom.}} \odot \boldsymbol{A}_i \boldsymbol{h}, \boldsymbol{H}_{\mathrm{het.}} \odot \boldsymbol{A}_i\right) \hat{\boldsymbol{W}}_i, \tag{19}$$

for $i = 1, \dots, n_b$, where the $\{\boldsymbol{W}_i \in \mathbb{R}^{n_a \times n_{out}}\}_{i=1}^{n_b}$ are the model parameters, and cat denotes the concatenation operator, $n_{out}$ is the expected output dimension. Then the output of this convolutional layer is the sum of all edge types

$$\hat{\boldsymbol{h}} = \sum_{i=1}^{n_b} \hat{\boldsymbol{h}}_i \in \mathbb{R}^{n \times n_{out}}. \tag{20}$$

In conclusion, given the input $\boldsymbol{h} \in \mathbb{R}^{n \times n_a}$ and bond tensor $\boldsymbol{E} \in \mathbb{R}^{n \times n \times n_b}$, the convolutional layer generates ouput

$$\text{GNN}^{\Gamma}(\boldsymbol{h} \mid \boldsymbol{E}) = \hat{\boldsymbol{h}} \in \mathbb{R}^{n \times n_{out}}. \tag{21}$$

### A.6 Computational Considerations

For convenience on the calculation of the log-likelihood, every transformation of variables needs the calculation of a Jacobian matrix (*i.e.*, $\partial \boldsymbol{Z}^{(l+1)}/\partial \boldsymbol{Z}^{(l)}$). So all the complicated modules (*e.g.*, GNNs, MLPs) are all built inside the coupling structure (part of the input is updated by the scaling matrix $\boldsymbol{S}$, and transformation matrix $\boldsymbol{T}$ depends on the other part of the input).

### A.7 Proof of Reversibility of HetFlows

Both the atom and bond models of HetFlows rely on ACL blocks. As introduced in App. A.1, these blocks are inherently reversible. This means they can forward process the input to produce an output and can also take that output to revert it to the original input without loss of information. Besides the use of ACL blocks, the operations used within the model primarily leverage simple concatenation or permutation. These operations are straightforward and do not affect the overall reversibility of the processes. Given that the individual components (both atom and bond flows) are reversible and the operations performed on the data are straightforward, thus that HetFlows as a whole is reversible. Fromal proof of the model's reversibility is provided below.

#### A.7.1 Reversibility of the ACL

**Claim** The affine coupling layer (ACL) defined at App. A.1, the atom model $f_a$ and bond model $f_b$ defined at App. A.2 are reversible. **Set up** Assume an ACL (Kingma & Dhariwal, 2018) defined in App. A.1 contains coupling function $f$ and masking matrix $\boldsymbol{M} \in \{0,1\}^{m \times n}$. Given input $\boldsymbol{X} \in \mathbb{R}^{m \times n}$, the output $\boldsymbol{Y}$ is calculated as

$$\begin{aligned} \boldsymbol{Y} &= \text{ACL}^{(f,\boldsymbol{M})}(\boldsymbol{X}) \\ &= \boldsymbol{M} \odot (\boldsymbol{S} \odot \boldsymbol{X}_1 + \boldsymbol{T}) + (\boldsymbol{1} - \boldsymbol{M})\boldsymbol{X}_2 \end{aligned} \tag{22}$$

where $\log \boldsymbol{S}, \boldsymbol{T} = f(\boldsymbol{X}_2)$, and $\boldsymbol{X}_1, \boldsymbol{X}_2$ are the split from input by masking:

$$\boldsymbol{X}_1 = \boldsymbol{M} \odot \boldsymbol{X}, \quad \boldsymbol{X}_2 = (\boldsymbol{1} - \boldsymbol{M}) \odot \boldsymbol{X}. \tag{23}$$

We seek to recover $\boldsymbol{X}$ from the $f, \boldsymbol{M}$, and $\boldsymbol{Y}$.

**Reversibility from output to input** Since $\boldsymbol{M}$ is binary, we can get the following results.

$$\boldsymbol{M} \odot \boldsymbol{M} = \boldsymbol{M}, \tag{24}$$
$$(\boldsymbol{1} - \boldsymbol{M}) \odot (\boldsymbol{1} - \boldsymbol{M}) = (\boldsymbol{1} - \boldsymbol{M}), \tag{25}$$
$$\boldsymbol{M} \odot (\boldsymbol{1} - \boldsymbol{M}) = (\boldsymbol{1} - \boldsymbol{M}) \odot \boldsymbol{M} = \boldsymbol{0} \tag{26}$$

and

$$\boldsymbol{X} = (\boldsymbol{M} + (\boldsymbol{1} - \boldsymbol{M})) \odot \boldsymbol{X} \tag{27}$$
$$= \boldsymbol{M} \odot \boldsymbol{X} + (\boldsymbol{1} - \boldsymbol{M}) \odot \boldsymbol{X} \tag{28}$$
$$= \boldsymbol{X}_1 + \boldsymbol{X}_2. \tag{29}$$

By splitting the output $\boldsymbol{Y}$ to $\boldsymbol{Y}_1, \boldsymbol{Y}_2$ by masking matrix:

$$\boldsymbol{Y}_1 = \boldsymbol{M} \odot \boldsymbol{Y}, \quad \boldsymbol{Y}_2 = (\boldsymbol{1} - \boldsymbol{M}) \odot \boldsymbol{Y}. \tag{30}$$

Combining with Eq. (22), we know

$$\boldsymbol{Y}_1 = \boldsymbol{M} \odot \boldsymbol{Y} \tag{31}$$
$$= \boldsymbol{M} \odot (\boldsymbol{M} \odot (\boldsymbol{S} \odot \boldsymbol{X}_1 + \boldsymbol{T}) + (\boldsymbol{1} - \boldsymbol{M}) \odot \boldsymbol{X}_2) \tag{32}$$
$$= \boldsymbol{M} \odot (\boldsymbol{S} \odot \boldsymbol{X}_1 + \boldsymbol{T}), \tag{33}$$

and

$$\boldsymbol{Y}_2 = (\boldsymbol{1} - \boldsymbol{M}) \odot \boldsymbol{Y} \tag{34}$$
$$= (\boldsymbol{1} - \boldsymbol{M}) \odot (\boldsymbol{M} \odot (\boldsymbol{S} \odot \boldsymbol{X}_1 + \boldsymbol{T})$$
$$+ (\boldsymbol{1} - \boldsymbol{M}) \odot \boldsymbol{X}_2) \tag{35}$$
$$= (\boldsymbol{1} - \boldsymbol{M}) \odot (\boldsymbol{M} \odot (\boldsymbol{S} \odot \boldsymbol{X}_1 + \boldsymbol{T})$$
$$+ (\boldsymbol{1} - \boldsymbol{M}) \odot (\boldsymbol{1} - \boldsymbol{M}) \odot \boldsymbol{X}) \tag{36}$$
$$= (\boldsymbol{1} - \boldsymbol{M}) \odot \boldsymbol{X} = \boldsymbol{X}_2. \tag{37}$$

Now the $\log \boldsymbol{S}, \boldsymbol{T} = f(\boldsymbol{X}_2) = \boldsymbol{Y}_2$ are recovered by $\boldsymbol{Y}$. Notice that

$$\boldsymbol{M} \odot (\boldsymbol{Y}_1 - \boldsymbol{T}) \oslash \boldsymbol{S} \tag{38}$$
$$= \boldsymbol{M} \odot (\boldsymbol{M} \odot (\boldsymbol{S} \odot \boldsymbol{X}_1 + \boldsymbol{T}) - \boldsymbol{T}) \oslash \boldsymbol{S} \tag{39}$$
$$= (\boldsymbol{M} \odot \boldsymbol{S} \odot \boldsymbol{X}_1 + \boldsymbol{M} \odot \boldsymbol{T} - \boldsymbol{M} \odot \boldsymbol{T}) \oslash \boldsymbol{S} \tag{40}$$
$$= (\boldsymbol{M} \odot \boldsymbol{S} \odot \boldsymbol{X}_1) \oslash \boldsymbol{S} \tag{41}$$
$$= \boldsymbol{M} \odot \boldsymbol{X}_1 \tag{42}$$
$$= \boldsymbol{M} \odot \boldsymbol{M} \odot \boldsymbol{X} \tag{43}$$
$$= \boldsymbol{M} \odot \boldsymbol{X} \tag{44}$$
$$= \boldsymbol{X}_1 \quad \text{if} \quad (\boldsymbol{S})_{i,j} > 0, \quad \forall i, j, \tag{45}$$

where '$\oslash$' denotes element-wise division. Since we define $\boldsymbol{S}$ as the exponential part of the output from the coupling function, the elements of $\boldsymbol{S}$ are all strictly positive. Then

$$\begin{aligned} \boldsymbol{X} &= \left( \mathrm{ACL}^{(f, \boldsymbol{M})} \right)^{-1} (\boldsymbol{Y}) \\ &= \boldsymbol{X}_1 + \boldsymbol{X}_2 \\ &= \boldsymbol{M} \odot (\boldsymbol{Y}_1 - \boldsymbol{T}) \oslash \boldsymbol{S} + \boldsymbol{Y}_2 \\ &= \boldsymbol{M} \odot (\boldsymbol{M} \odot \boldsymbol{Y} - \boldsymbol{T}) \oslash \boldsymbol{S} + (\boldsymbol{1} - \boldsymbol{M}) \odot \boldsymbol{Y}. \end{aligned} \tag{46}$$

where $\log \boldsymbol{S}, \boldsymbol{T} = f(\boldsymbol{X}_2) = f((\boldsymbol{1} - \boldsymbol{M}) \odot \boldsymbol{Y})$. Eq. (46) shows how the input is recovered from the output, thus the ACL block is reversible.

### A.7.2 Reversibility of the Bond Model

For the bond model $f_b = \mathrm{ACL}_{k_b}^b \circ \cdots \circ \mathrm{ACL}_1^b$, and since each $\mathrm{ACL}_i^b$, $i = 1, \ldots, k_b$ is reversible, we can write $f_b^{-1} = \left( \mathrm{ACL}_1^b \right)^{-1} \circ \cdots \circ \left( \mathrm{ACL}_{k_b}^b \right)^{-1}$, which the reverse function of $f_b$.

### A.7.3 Reversibility of the Atom Model

For the atom model $f_a = \mathrm{ACL}_{k_a}^a \circ \cdots \circ \mathrm{ACL}_1^a$, and since each $\mathrm{ACL}_i^a$, $i = 1, \ldots, k_a$ is reversible, we can write $f_a^{-1} = \left( \mathrm{ACL}_1^a \right)^{-1} \circ \cdots \circ \left( \mathrm{ACL}_{k_a}^a \right)^{-1}$, which the reverse function of $f_a$.

## B   Experiment Details: Node Classification

### B.1   Hardware

All models in this experiment are trained on a Linux cluster equipped with NVIDIA V100 GPUs. The training time and memory requirement for single were (for all modes orig., hom., het., mix. and for all architectures):

- TEXAS: 10 mins, 2 GB

- CORNELL: 10 mins, 2 GB

- WISCONSIN: 10 mins, 2 GB

- SQUIRREL: 15 mins, 16 GB

- CHAMELEON 10 mins, 4 GB

- CITESEER: 10 mins, 2 GB

- COMPUTERS: 20 mins, 16 GB

- PUBMED: 10 mins, 4 GB

- CORA: 10 mins, 2 GB

- PHOTO: 15 mins, 8 GB

- ROMAN-EMPIRE: 10 mins, 2GB

- AMAZON-RATINGS: 10 mins, 2GB

- MINESWEEPER: 10 mins, 2GB

- TOLOKERS: 20 mins, 16GB

- QUESTIONS: 15 mins, 8GB

## B.2 Data Sets

There are 15 data sets selected to conduct comprehensive and discriminative node classification tasks. To examine patterns associated with varying levels of data homophily, half of the data sets are chosen as having higher homophily, while the other half are more heterophilic. The data domains include citation networks (Yang et al., 2016) (CORA, PUBMED, CITESEER), co-purchase graphs (Shchur et al., 2018) (COMPUTERS, PHOTO), hyperlink networks (Pei et al., 2019) (CORNELL, WISCONSIN, TEXAS), Wikipedia networks (Rozemberczki et al., 2021) (CHAMELEON, SQUIRREL), and heterophilous graph dataset (Platonov et al., 2023) (ROMAN-EMPIRE, AMAZON-RATINGS, MINESWEEPER, TOLOKERS, QUESTIONS).

**Statistical information**  The descriptive statistics on all the data sets for node classification tasks in Sec. 4.1 are displayed in Table A4. Here, $N_{\text{graphs}}, N_{\text{nodes}}$ and $N_{\text{edges}}$ denote the total amounts of graphs, nodes and edges in the data set respectively. $D_{\text{feat}}, N_{\text{class}}$ denote the dimensions of the node features and labels. $\mathcal{H}_n$, $\mathcal{H}_e$ and $\mathcal{H}_{ei}$ denote the average node homophily (Pei et al., 2019), average edge homophily (Zhu et al., 2020) and average class insensitive edge homophily ratio (Lim et al., 2021) of the data set.

**Data split rule**  For three heterophilic data sets (CORNELL, WISCONSIN and TEXAS), the data is split to train/validate/test with fixed 10 seeds from GEOM-GCN (Pei et al., 2019). For other datasets, the splitting is randomly controlled by the python *torch* package. Each result contains 10 random runs for data split and model initializations.

## B.3 GNN Algorithm Details

There are four classic GNNs utilized as base models in the node classification task of Sec. 4.1. Here we provide these algorithms by showing the message-passing details of a single convolutional layer.

**Graph Convolutional Networks (GCN)**  The most classic graph convolutional operator is proposed by Kipf & Welling (2017). Given the embeddings $\boldsymbol{h}_v^{(k)}$ of node $v \in \mathcal{V}$ at $k^{\text{th}}$ layer, the message for node $u$ from node $v$ is defined as

$$\boldsymbol{m}_{uv}^{(k)} = \frac{e_{u,v}}{(\hat{d}_u \hat{d}_v)^{1/2}} \boldsymbol{h}_v^{(k)}, \tag{47}$$

where $e_{u,v}$ denotes the weight of edge $(u, v)$, $\hat{d}_v = 1 + \sum_{u \in \mathcal{N}(v)} e_{u,v}$ denotes the weighted degree of node $v$ with self-loop. Then the node embeddings are updated by the message set $\{\{\boldsymbol{m}_{uv} | v \in \mathcal{N}(u) \cup \{u\}\}\}$ collected from all its neighbours

$$\boldsymbol{h}_u^{(k+1)} = \Theta^\top \sum_{u \in \mathcal{N}(u) \cup \{u\}} \boldsymbol{m}_{uv}^{(k)}, \tag{48}$$

where $\Theta$ denotes the model parameters.

**Graph Attention Networks (GAT)**   Veličković et al. (2018) proposed the graph attention operator inspired by the transformer. Given the embeddings $\boldsymbol{h}_v^{(k)}$ of node $v \in \mathcal{V}$ at $k^{\text{th}}$ layer, the message for node $v$ from node $u$ is defined as

$$\boldsymbol{m}_{uv}^{(k)} = \begin{cases} \alpha_{u,v} \Theta_t \boldsymbol{h}_u^{(k)} & \text{if } u \neq v, \\ \alpha_{u,v} \Theta_s \boldsymbol{h}_v^{(k)} & \text{if } u = v \end{cases}, \tag{49}$$

where $\Theta_s, \Theta_t$ denotes the model parameters, and $\alpha_{u,v}$ denotes the attention value from node $v$ to node $u$

$$\alpha_{u,v} = \frac{\exp\left(\text{LR}(\boldsymbol{a}_s^\top \Theta_s \boldsymbol{h}_u + \boldsymbol{a}_t^\top \Theta_t \boldsymbol{h}_v)\right)}{\sum\limits_{w \in \mathcal{N}(u) \cup \{u\}} \exp\left(\text{LR}(\boldsymbol{a}_s^\top \Theta_s \boldsymbol{h}_u + \boldsymbol{a}_t^\top \Theta_t \boldsymbol{h}_w)\right)}. \tag{50}$$

where LR denotes the Leaky Rectified Linear Unit. Then the node embeddings are updated by the neighbour message set

$$\boldsymbol{h}_u^{(k+1)} = \sum_{u \in \mathcal{N}(u) \cup \{u\}} \boldsymbol{m}_{uv}^{(k)}. \tag{51}$$

**Graph Isomorphism Network (GIN)**   The graph isomorphism operator is created by Xu et al. (2019) based on Weisfeiler–Lehman (WL) graph isomorphism test (Leman & Weisfeiler, 1968). Given the embeddings $\boldsymbol{h}_v^{(k)}$ of node $v \in \mathcal{V}$ at $k^{\text{th}}$ layer, the message for node $v$ from node $u$ is the node embedding itself

$$\boldsymbol{m}_{uv}^{(k)} = \begin{cases} \boldsymbol{h}_v^{(k)} & \text{if } u \neq v \\ (1 + \varepsilon) \boldsymbol{h}_u^{(k)} & \text{if } u = v \end{cases}, \tag{52}$$

where the $\varepsilon$ controls the self-weight. Then the node embeddings are updated by the neighbour message set through another neural network $h_\theta$

$$\boldsymbol{h}_u^{(k+1)} = h_\theta \left( \sum_{u \in \mathcal{N}(u) \cup \{u\}} \boldsymbol{m}_{uv}^{(k)} \right). \tag{53}$$

Table A4: The statistic information on the data sets for node classification experiment in Sec. 4.1

| | $N_{\text{graphs}}$ | $N_{\text{nodes}}$ | $N_{\text{edges}}$ | $D_{\text{feat}}$ | $N_{\text{class}}$ | $\mathcal{H}_n$ | $\mathcal{H}_e$ | $\mathcal{H}_{ei}$ |
|---|---|---|---|---|---|---|---|---|
| TEXAS | 1 | 183 | 325 | 1703 | 5 | 0.104 | 0.108 | 0.001 |
| MINESWEEPER | 1 | 10000 | 78804 | 7 | 2 | 0.683 | 0.683 | 0.009 |
| ROMAN-EMPIRE | 1 | 22662 | 65854 | 300 | 18 | 0.046 | 0.047 | 0.021 |
| SQUIRREL | 1 | 5201 | 217073 | 2089 | 5 | 0.219 | 0.224 | 0.026 |
| CORNELL | 1 | 183 | 298 | 1703 | 5 | 0.106 | 0.131 | 0.059 |
| CHAMELEON | 1 | 2277 | 36101 | 2325 | 5 | 0.249 | 0.235 | 0.063 |
| QUESTIONS | 1 | 48921 | 307080 | 301 | 2 | 0.898 | 0.840 | 0.079 |
| WISCONSIN | 1 | 251 | 515 | 1703 | 5 | 0.134 | 0.196 | 0.097 |
| AMAZON-RATINGS | 1 | 24492 | 186100 | 300 | 5 | 0.376 | 0.380 | 0.127 |
| TOLOKERS | 1 | 11758 | 1038000 | 10 | 2 | 0.634 | 0.595 | 0.180 |
| CITESEER | 1 | 3327 | 9104 | 3703 | 6 | 0.706 | 0.736 | 0.627 |
| PUBMED | 1 | 19717 | 88648 | 500 | 3 | 0.792 | 0.802 | 0.664 |
| COMPUTERS | 1 | 13752 | 491722 | 767 | 10 | 0.785 | 0.777 | 0.700 |
| CORA | 1 | 2708 | 10556 | 1433 | 7 | 0.825 | 0.810 | 0.766 |
| PHOTO | 1 | 7650 | 238162 | 745 | 8 | 0.836 | 0.827 | 0.772 |

Table A5: Performance comparison between GraphSAGE+mix. with other heterophily-aware GNNs, the best result among the baseline modes is highlighted in bold.

| Homophily $\mathcal{H}_{ei}$ | Texas 0.00 | Cornell 0.06 | Wisconsin 0.10 | CiteSeer 0.63 | PubMed 0.66 | Cora 0.77 |
|---|---|---|---|---|---|---|
| GraphSAGE+mix. | $79.7_{\pm5.9}$ | $75.1_{\pm4.0}$ | $84.5_{\pm1.1}$ | $77.0_{\pm1.0}$ | $89.3_{\pm0.4}$ | $87.8_{\pm1.2}$ |
| GraphSAGE | $79.0_{\pm1.2}$ | $71.4_{\pm1.2}$ | $64.8_{\pm5.1}$ | $78.2_{\pm0.3}$ | $86.8_{\pm0.1}$ | $86.6_{\pm0.3}$ |
| ACMP | $86.2_{\pm3.0}$ | $85.4_{\pm7.0}$ | $86.1_{\pm4.0}$ | $75.0_{\pm1.0}$ | $78.9_{\pm1.0}$ | $84.9_{\pm0.6}$ |
| H2GCN | $85.9_{\pm3.5}$ | $86.2_{\pm4.7}$ | $87.5_{\pm1.8}$ | $80.0_{\pm0.7}$ | $87.8_{\pm0.3}$ | $87.5_{\pm0.6}$ |
| GPRGNN | $92.9_{\pm0.6}$ | $91.4_{\pm0.7}$ | $93.8_{\pm2.4}$ | $67.6_{\pm0.4}$ | $85.1_{\pm0.1}$ | $79.5_{\pm0.4}$ |
| Geom-GCN | $67.6_{\pm N/A}$ | $60.8_{\pm N/A}$ | $64.1_{\pm N/A}$ | $78.0_{\pm N/A}$ | $90.0_{\pm N/A}$ | $85.3_{\pm N/A}$ |
| ACM-GCN | $\mathbf{94.9}_{\pm2.9}$ | $\mathbf{94.8}_{\pm3.8}$ | $\mathbf{95.8}_{\pm2.0}$ | $\mathbf{81.7}_{\pm1.0}$ | $\mathbf{90.7}_{\pm0.5}$ | $\mathbf{88.6}_{\pm1.2}$ |

The $h_\theta$ is often defined as a simple MLP.

**GraphSAGE** Hamilton et al. (2017) highlights the sample and aggregation approach in their GraphSAGE operator. Given the embeddings $\boldsymbol{h}_v^{(k)}$ of node $v \in \mathcal{V}$ at $k^{\text{th}}$ layer, the message for node $v$ from node $u$ is the node embedding itself

$$\boldsymbol{m}_{uv}^{(k)} = vh_v^{(k)}u, \tag{54}$$

Then the node embeddings are updated as

$$\boldsymbol{h}_u^{(k+1)} = \Theta_1 \boldsymbol{m}_{uu} + \Theta_2 \frac{1}{|\mathcal{N}(u)|} \sum_{v \in \mathcal{N}(u)} \boldsymbol{m}_{uv}, \tag{55}$$

where $\Theta_1, \Theta_2$ are the model parameters.

### B.4 Comparison with other heterophily-aware GNN baselines

We select GraphSAGE+mix. to compare with baselines on node classification tasks over 6 data sets in Table A5. The baseline models are selected as the base model and other heterophily-aware GNN: ACMP (Wang et al., 2023), H2GCN (Zhu et al., 2021), GPRGNN (Chien et al., 2021), Geom-GCN (Pei et al., 2019) and ACM-GCN (Luan et al., 2022). The GraphSAGE+mix. are generally better than its original model: GraphSAGE. It shows the representation enhancement through our heterophilous message-passing scheme. Here ACM-GCN is the best algorithm over all algorithms in such comparison. The N/A means the reported results are not available from the baseline papers.

## C Experiment Details: Molecule Generation

### C.1 Hardware

All models in this experiment are trained on a cluster equipped with NVIDIA A100 GPUs. The training time for a single model was 10 h (QM9) and 90 h (ZINC-250K).

The training time and memory requirement for single models were (for all modes orig., hom.or het.and for all base models)

- QM9: 10 h, 8 GB

- ZINC-250K: 90 h, 16 GB

### C.2 Data sets

Two classic molecule data sets (QM9 and ZINC-250K), which are widely used in academia, are chosen for the molecular generation task. The QM9 data set (Ramakrishnan et al., 2014) comprises ~134k stable small organic molecules composed of atoms from the set {C, H, O, N, F}. The ZINC-250K (Irwin et al., 2012) data contains ~250k drug-like molecules, each with up to 38 atoms of 9 different types.

**Data split** In this experiment, all data sets are split with ratio train/test = 80/20%.

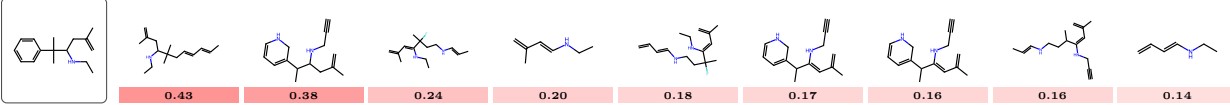

Figure A6: **Structured latent-space exploration (**ZINC-250K**).** Example of nearest neighbour search in the latent space with the seed molecules on the left and neighbours with the Tanimoto similarity (1 ▮▮▮▮ 0) given for each molecule. For results on QM9, see Fig. 4 in the main paper.

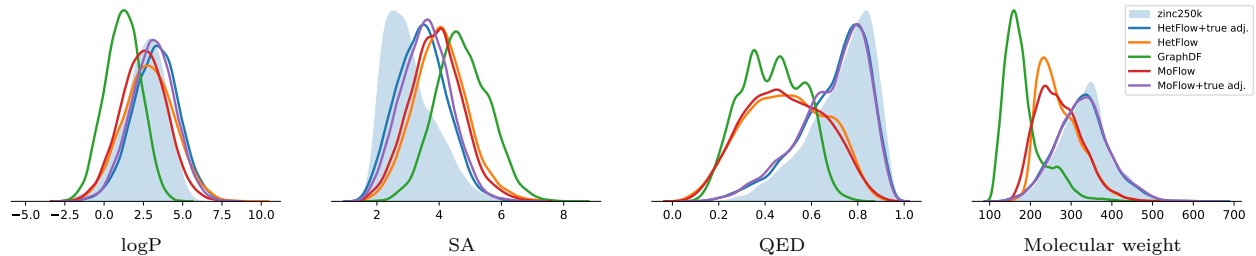

Figure A7: Chemoinformatics statistics for data (ZINC-250K) and generated molecules from HetFlows (ours), MoFlow, and GraphDF. Histograms for the Octanol-water partition coefficient (logP), synthetic accessibility score (SA), quantitative estimation of drug-likeness (QED), and molecular weight.

Table A6: Full benchmark metrics for random generation using QM9 (reporting mean±std)

| | | Validity ↑ | Uniqueness ↑ | Novelty ↑ | SNN ↑ | Frag ↑ | Scaf ↑ | IntDiv$_1$ ↑ |
|---|---|---|---|---|---|---|---|---|
| | Data (QM9) | 1.00±0.00 | 1.00±0.00 | 0.62±0.02 | 0.54±0.00 | 0.94±0.01 | 0.76±0.03 | 0.92±0.00 |
| Flows | GraphDF | - | **1.00**±0.00 | **0.98**±0.00 | 0.35±0.00 | 0.61±0.01 | 0.09±0.07 | 0.87±0.00 |
| | MoFlow | 0.95±0.01 | **1.00**±0.00 | 0.96±0.01 | 0.32±0.00 | 0.60±0.03 | 0.04±0.03 | **0.92**±0.00 |
| | HetFlow (Ours) | 0.92±0.01 | 0.99±0.00 | 0.92±0.01 | 0.34±0.00 | 0.80±0.02 | 0.04±0.03 | 0.91±0.00 |
| Abl. | MoFlow+true adj. | **1.00**±0.00 | **1.00**±0.00 | 0.85±0.01 | 0.38±0.00 | 0.70±0.03 | 0.31±0.08 | **0.92**±0.00 |
| | HetFlow+true adj. | **1.00**±0.00 | **1.00**±0.00 | 0.74±0.01 | 0.43±0.00 | 0.85±0.02 | 0.52±0.05 | **0.92**±0.00 |
| Diff. | EDM | 0.92±0.01 | **1.00**±0.00 | 0.56±0.02 | **0.48**±0.01 | **0.92**±0.01 | **0.65**±0.03 | **0.92**±0.00 |

| | | IntDiv$_2$ ↑ | Filters ↑ | FCD ↓ | ΔlogP ↓ | ΔSA ↓ | ΔQED ↓ | ΔWeight ↓ |
|---|---|---|---|---|---|---|---|---|
| | Data (QM9) | 0.90±0.00 | 0.64±0.02 | 0.40±0.02 | 0.04±0.01 | 0.03±0.01 | 0.00±0.00 | 0.32±0.08 |
| Flows | GraphDF | 0.86±0.00 | **0.69**±0.02 | 10.76±0.21 | 0.16±0.03 | 0.27±0.02 | 0.05±0.00 | 19.72±0.54 |
| | MoFlow | 0.90±0.00 | 0.55±0.02 | 7.55±0.23 | 0.40±0.02 | 0.41±0.02 | 0.04±0.00 | 3.75±0.08 |
| | HetFlow (Ours) | 0.90±0.00 | 0.62±0.02 | 4.04±0.24 | **0.11**±0.02 | 0.34±0.01 | 0.03±0.00 | 4.40±0.11 |
| Abl. | MoFlow+true adj. | 0.90±0.00 | 0.60±0.01 | 4.45±0.11 | 0.65±0.02 | 0.56±0.02 | 0.04±0.00 | **0.74**±0.14 |
| | HetFlow+true adj. | 0.90±0.00 | 0.62±0.01 | 1.46±0.09 | 0.26±0.02 | 0.36±0.02 | 0.02±0.00 | 1.12±0.18 |
| Diff. | EDM | 0.90±0.00 | 0.61±0.01 | **0.96**±0.08 | 0.16±0.04 | **0.15**±0.04 | **0.01**±0.00 | 1.87±0.16 |

## C.3   Further Results

**Latent-space exploration**   We provide further results for structured latent-space exploration. Example explorations for QM9 and ZINC-250K are shown in Fig. 4 and Fig. A6.

**Overall 14 metrics tables**   We include full listings of all 14 metrics (description of metrics in App. D) considered in the random generation tasks for QM9 and ZINC-250K. The values are listed in Tables A6 and A7, respectively.

**The reconstruction example**   For better understanding, we provide reconstruction examples on QM9 from intermediate layers in Fig. A8.

Table A7: Full benchmark metrics for random generation using ZINC-250K (reporting mean±std)

| | | Validity ↑ | Uniqueness ↑ | Novelty ↑ | SNN ↑ | Frag ↑ | Scaf ↑ | IntDiv$_1$ ↑ |
|---|---|---|---|---|---|---|---|---|
| | Data (ZINC-250K) | $1.00_{\pm0.00}$ | $1.00_{\pm0.00}$ | $0.02_{\pm0.00}$ | $0.51_{\pm0.00}$ | $1.00_{\pm0.00}$ | $0.28_{\pm0.02}$ | $0.87_{\pm0.00}$ |
| Flows | GraphDF | - | $\mathbf{1.00}_{\pm0.00}$ | $\mathbf{1.00}_{\pm0.00}$ | $0.23_{\pm0.00}$ | $0.35_{\pm0.01}$ | $0.00_{\pm0.00}$ | $\mathbf{0.88}_{\pm0.00}$ |
| | MoFlow | $0.89_{\pm0.01}$ | $\mathbf{1.00}_{\pm0.00}$ | $\mathbf{1.00}_{\pm0.00}$ | $0.27_{\pm0.00}$ | $0.79_{\pm0.01}$ | $0.01_{\pm0.00}$ | $\mathbf{0.88}_{\pm0.00}$ |
| | HetFlow (Ours) | $0.87_{\pm0.01}$ | $\mathbf{1.00}_{\pm0.00}$ | $\mathbf{1.00}_{\pm0.00}$ | $0.26_{\pm0.00}$ | $0.77_{\pm0.01}$ | $0.01_{\pm0.00}$ | $\mathbf{0.88}_{\pm0.00}$ |
| Abl. | MoFlow+true adj. | $\mathbf{0.94}_{\pm0.01}$ | $\mathbf{1.00}_{\pm0.00}$ | $\mathbf{1.00}_{\pm0.00}$ | $0.33_{\pm0.00}$ | $0.89_{\pm0.01}$ | $0.07_{\pm0.02}$ | $0.87_{\pm0.00}$ |
| | HetFlow+true adj. | $0.93_{\pm0.01}$ | $\mathbf{1.00}_{\pm0.00}$ | $\mathbf{1.00}_{\pm0.00}$ | $\mathbf{0.34}_{\pm0.00}$ | $\mathbf{0.91}_{\pm0.01}$ | $\mathbf{0.10}_{\pm0.03}$ | $0.87_{\pm0.00}$ |

| | | IntDiv$_2$ ↑ | Filters ↑ | FCD ↓ | ΔlogP ↓ | ΔSA ↓ | ΔQED ↓ | ΔWeight ↓ |
|---|---|---|---|---|---|---|---|---|
| | Data (ZINC-250K) | $0.86_{\pm0.00}$ | $0.59_{\pm0.01}$ | $1.44_{\pm0.01}$ | $0.05_{\pm0.01}$ | $0.03_{\pm0.01}$ | $0.01_{\pm0.00}$ | $2.18_{\pm0.39}$ |
| Flows | GraphDF | $\mathbf{0.87}_{\pm0.00}$ | $0.54_{\pm0.01}$ | $34.30_{\pm0.30}$ | $1.28_{\pm0.03}$ | $1.70_{\pm0.03}$ | $0.30_{\pm0.00}$ | $149.27_{\pm1.55}$ |
| | MoFlow | $0.86_{\pm0.00}$ | $0.51_{\pm0.02}$ | $23.33_{\pm0.35}$ | $\mathbf{0.15}_{\pm0.02}$ | $0.82_{\pm0.03}$ | $0.25_{\pm0.01}$ | $56.71_{\pm2.64}$ |
| | HetFlow (Ours) | $\mathbf{0.87}_{\pm0.00}$ | $0.51_{\pm0.01}$ | $23.72_{\pm0.19}$ | $0.50_{\pm0.04}$ | $0.99_{\pm0.03}$ | $0.25_{\pm0.01}$ | $51.90_{\pm1.65}$ |
| Abl. | MoFlow+true adj. | $0.86_{\pm0.00}$ | $0.67_{\pm0.02}$ | $\mathbf{8.21}_{\pm0.22}$ | $0.64_{\pm0.04}$ | $0.54_{\pm0.03}$ | $\mathbf{0.04}_{\pm0.00}$ | $\mathbf{4.23}_{\pm0.51}$ |
| | HetFlow+true adj. | $0.86_{\pm0.00}$ | $\mathbf{0.75}_{\pm0.01}$ | $8.24_{\pm0.17}$ | $0.82_{\pm0.04}$ | $0.41_{\pm0.03}$ | $\mathbf{0.04}_{\pm0.00}$ | $5.05_{\pm0.77}$ |

| Layer $i{=}0$ | Layer $i{=}4$ | Layer $i{=}8$ | Layer $i{=}10$ | Layer $i{=}16$ | Layer $i{=}20$ | Layer $i{=}26$ |

Figure A8: **Step-by-step generation (**QM9**).** Snapshots of reconstructed molecules when fixing the bond model and collecting node embeddings of the intermediate layers $i$.

**Homophily distribution of generated molecules**   Additionally, we also visualize the node homophily for both QM9 and ZINC-250K together with the estimated node homophily histograms (see Fig. A9) from the generation outputs from the different models. The adjacency matrix generation strategies have a minor influence on the homophily distribution of generated molecules, it shows these statistics mostly rely on the atom model but not the bond model.

**Ablation study**   The random generation of HetFlows with/without parameters sharing in the MixMP GNNs are shown in Table A8 and Table A9.

Table A8: Ablation study of MixMP GNN parameter sharing on QM9

| | FCD ↓ | Validity ↑ | Novelty ↑ | SNN ↑ | Frag ↑ | Scaf ↑ | IntDiv$_1$ ↑ |
|---|---|---|---|---|---|---|---|
| Data (QM9) | $0.40_{\pm0.02}$ | $1.00_{\pm0.00}$ | $0.62_{\pm0.02}$ | $0.54_{\pm0.00}$ | $0.94_{\pm0.01}$ | $0.76_{\pm0.03}$ | $0.92_{\pm0.00}$ |
| HetFlow+true adj. | $\mathbf{1.46}_{\pm0.09}$ | $1.00_{\pm0.00}$ | $0.74_{\pm0.01}$ | $\mathbf{0.43}_{\pm0.00}$ | $0.85_{\pm0.02}$ | $\mathbf{0.52}_{\pm0.05}$ | $0.92_{\pm0.00}$ |
| HetFlow+true adj.+share para. | $1.66_{\pm0.07}$ | $1.00_{\pm0.00}$ | $0.77_{\pm0.01}$ | $0.42_{\pm0.00}$ | $\mathbf{0.86}_{\pm0.02}$ | $0.48_{\pm0.05}$ | $0.92_{\pm0.00}$ |

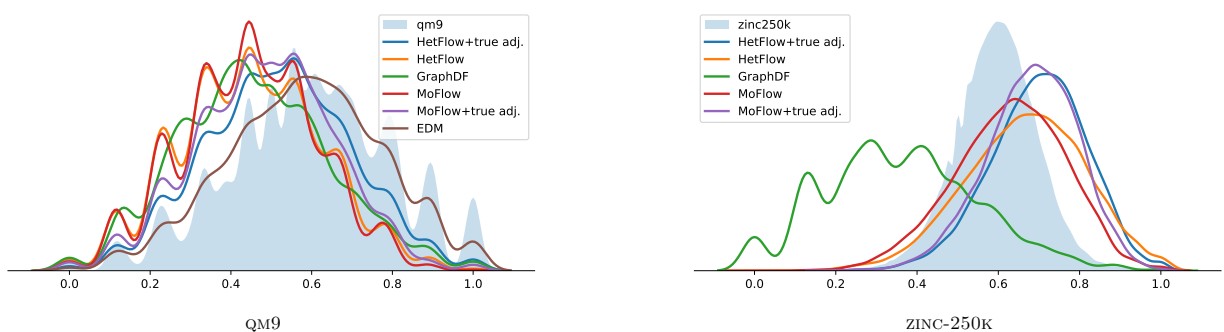

Figure A9: Node homophily distribution of generated molecules.

Table A9: Ablation study of MixMP GNN parameter sharing on ZINC-250K

| | FCD ↓ | Validity ↑ | Novelty ↑ | SNN ↑ | Frag ↑ | Scaf ↑ | IntDiv$_1$ ↑ |
|---|---|---|---|---|---|---|---|
| Data (ZINC-250K) | 1.44±0.01 | 1.00±0.00 | 0.02±0.00 | 0.51±0.00 | 1.00±0.00 | 0.28±0.02 | 0.87±0.00 |
| HetFlow+true adj. | 8.24±0.17 | **0.93**±0.01 | **1.00**±0.00 | **0.34**±0.00 | 0.91±0.01 | **0.10**±0.03 | **0.87**±0.00 |
| HetFlow+true adj.+share para. | **8.11**±0.21 | **0.93**±0.01 | **1.00**±0.00 | 0.32±0.00 | **0.93**±0.01 | 0.05±0.02 | **0.87**±0.00 |

Table A10: Performance on molecule property optimization in terms of the best QED scores, scores taken from the corresponding papers (JTVAE score from Luo et al., 2021; Verma et al., 2022).

| Method | 1st | 2nd | 3rd |
|---|---|---|---|
| Data (ZINC-250K) | 0.948 | 0.948 | 0.948 |
| JTVAE | 0.925 | 0.911 | 0.910 |
| GCPN | **0.948** | 0.947 | 0.946 |
| GraphAF | **0.948** | **0.948** | 0.947 |
| GraphDF | **0.948** | **0.948** | **0.948** |
| MoFlow | **0.948** | **0.948** | **0.948** |
| ModFlow | **0.948** | **0.948** | 0.945 |
| HetFlows | **0.948** | **0.948** | 0.947 |

## C.4 Property Optimization

In the property optimization task, models show their capability to find novel molecules that optimize specific chemical properties not present in the training data set: a critical component for drug discovery. For our study, we focused on maximizing the QED property. We trained HetFlows on ZINC-250K and evaluated its performance against other state-of-the-art models (Verma et al., 2022; Luo et al., 2021; Zang & Wang, 2020; Shi et al., 2019; Jin et al., 2018; You et al., 2018). The results, given in Table A10, show that the top three novel molecule candidates identified by HetFlows (not part of the ZINC-250K data set), exhibit QED values on par with those from ZINC-250K or other state-of-the-art methods.

**Algorithm** Given a pre-trained HetFlows $f$, and training set $\mathcal{D}$ contains molecule and property label pairs $\{G, y\}$. Now we introduce an extra simple MLP $g_\theta$, which maps the molecular embeddings $f(G)$ into the predicted property

$$y_p = g_\theta(f(G)), \tag{56}$$

it is trained on the dataset to be $g_{\theta*}$ by optimizing the parameters:

$$\theta^* = \arg\min_\theta \operatorname{MSEloss}_{(G,y)\in\mathcal{D}}(g_\theta(f(G)), y) \tag{57}$$

Then we find molecule candidates $\{G_i\}_{i=1}^k$ with top-$k$ properties in the data set $\mathcal{D}$ are chosen. New embeddings are explored by optimizing the predict label by $g_{\theta*}$ starting from these candidates:

$$
\begin{aligned}
\boldsymbol{h}_{i,j} &= \delta \frac{\partial g_{\theta*}}{\partial \boldsymbol{h}}(\boldsymbol{h}_{i,j-1}) + \boldsymbol{h}_{i,j-1}, \quad j = 1, \dots, N, \\
\boldsymbol{h}_{i,0} &= f(G_i), \quad i = 1, \dots, k,
\end{aligned}
\tag{58}
$$

where $\delta$ denotes the search step length, and $N$ is the number of iterations. These embeddings could be recovered to be molecule set:

$$
\mathcal{D}' = \{f^{-1}(\boldsymbol{h}_{ij})\}_{i=1,\dots,k, \quad j=1,\dots,N}.
\tag{59}
$$

Finally, $\mathcal{D}'\backslash\mathcal{D}$ gives the novel molecule sets with related high target properties.

**Generation results**  In our experiments, the $g_\theta$ is a simple 3-layer MLP with 16 hidden nodes, the dataset $\mathcal{D}$ is ZINC-250K, and target property $y$ is QED. And $\mathcal{D}'\backslash\mathcal{D}$ provides 17 molecules with QED score 0.948. The Top-3 QED score and molecular SMILES are listed below:

1. Cc1cc(C(=O)NC2CC2)c(C)n1-c1ccc2c(c1)OCCO2.N, QED = 0.947936,

2. Cc1cccnc1NC(=O)C1CC(=O)N(C)C1c1ccccc1, QED = 0.947505,

3. Cc1nn(C)c(C)c1C(=O)NCC1CC12CCc1ccccc12, QED = 0.947317.

**Baselines**  The baselines scores of GCPN (You et al., 2018), GraphAF (Shi et al., 2019), GraphDF (Luo et al., 2021), MoFlow (Zang & Wang, 2020) and ModFlow (Verma et al., 2022) are acquired from the corresponding papers. The score of JTVAE (Jin et al., 2018) is acquired from Zang & Wang (2020); Verma et al. (2022).

### C.5 Model selection

The ranges of HetFlows hyperparameters are listed as follows:

- The residual connection of ACL coupling function: [True, False]

- The parameter sharing mode of $\ln \mu_a, \ln \mu_b$: $[0, 1, 2]$. Here $\mu_a, \mu_b$ are the variance of the embeddings distribution $\mathcal{N}_a, \mathcal{N}_b$ in Eq. (7).

  - mode 0 means the distribution variance of both node and edge embeddings are fixed to be 1: $\ln \mu_a = \ln \mu_b = 0$.
  - mode 1 means the distribution variance of both node and edge embeddings share one parameter: $\ln \mu_a = \ln \mu_b$, it could be optimized during the training process.
  - mode 2 means the distribution variance of both node and edge embeddings are separate parameters: $\ln \mu_a, \ln \mu_b$, both of them could be optimized the during training process.

The best-performing model is selected using the FCD score as suggested in Polykovskiy et al. (2020).

The HetFlows reported in the paper is selected with hyperparameters:

- The residual connection of ACL coupling function: False

- The parameter sharing mode of $\ln \mu_a, \ln \mu_b$: 1.

# D   Description of Chemoinformatics Metrics

For benchmarking, model selection, comparison, and explorative analysis, we use the following 14 metrics. The metrics are presented in detail in the work by Polykovskiy et al. (2020) that introduced the MOSES benchmarking platform. The metrics calculation makes heavy use of the RDKit open-source cheminformatics software (https://www.rdkit.org/). We briefly summarize the metrics below.

**Sanity check metrics**

1. **Validity** Fraction (in $[0, 1]$) of the molecules that produce valid SMILES representations. This is a sanity check for how well the model captures explicit chemical constraints such as proper valence. Higher values are better as a low value can indicate that the model does not properly capture chemical structure. We report numbers without *post hoc* validity corrections.

2. **Uniqueness** Fraction (in $[0, 1]$) of the molecules that are unique. This is a sanity check based on the SMILES string representation of the generated molecules. Higher values are better as a low value can indicate the model has collapsed and produces only a few typical molecules.

3. **Novelty** Fraction (in $[0, 1]$) of the generated molecules that are not present in the training set. Higher values are better as a low value can indicate overfitting to the training data set.

**Summary statistics**

4. **Similarity to a nearest neighbour (SNN)** The average Tanimoto similarity (Jaccard coefficient) in $[0, 1]$ between the generated molecules and their nearest neighbour in the reference data set. Higher is better: If the generated molecules are far from the reference set, similarity to the nearest neighbour will be low.

5. **Fragment similarity (Frag)** Measures similarity (in $[0, 1]$) of distributions of BRICS fragments (substructures) in the generated set vs. the original data set. If molecules in the two sets share many of the same fragments in similar proportions, the Frag metric will be close to 1 (higher better).

6. **Scaffold similarity (Scaf)** Measures similarity (in $[0, 1]$) of distributions of Bemis–Murcko scaffolds (molecule ring structures, linker fragments, and carbonyl groups) in the generated set vs. the original data set. This metric is calculated similarly to the Fragment similarity metric by counting substructure presence in the data, and they can be high even if the data sets do not contain the same molecules.

7. **Internal diversity (IntDiv$_1$)** Measure (in $[0, 1]$) of the chemical diversity within the generated set of molecules. Higher values are better and signal higher diversity in the generated set of molecules. Low values can signal mode collapse.

8. **Internal diversity (IntDiv$_2$)** Measure (in $[0, 1]$) of the chemical diversity within the generated set of molecules. The interpretation is similar to IntDiv$_1$ but with stronger penalization of the Tanimoto similarity in calculating the diversity.

9. **Filters** This metric is specific to the MOSES benchmarking platrofm (see Polykovskiy et al., 2020). It gives the fraction (in $[0, 1]$) of generated molecules that pass filters applied during data set construction. In practice, these filters may filter out chemically valid molecules with fragments that are not of interest in the MOSES data set (filtered with medicinal chemistry filters). Thus, this metric is not of primary interest to us but gives a view on the match with the MOSES data set.

10. **Fréchet ChemNet distance (FCD)** Analogous to the Frechét Inception Distance (FID) used in image generation, FCD compares feature distributions of real and generated molecules using a pre-trained model (ChemNet). Lower values are better.

**Descriptive distributions**

11. **Octanol-water partition coefficient (logP)** A logarithmic measure of the relationship between lipophilicity (fat solubility) and hydrophilicity (water solubility) of a set of molecules. For large values a substance is more soluble in fat-like solvents such as n-octanol, and for small values more soluble in water. We report both histograms of logP and a summary statistic in terms of the Wasserstein distance between the generated and reference distributions (smaller better).

12. **Synthetic accessibility score (SA)** A metric that estimates how easily a chemical molecule can be synthesized. It provides a quantitative value indicating the relative difficulty or ease of synthesizing a molecule, with a lower SA score suggesting that a molecule is more easily synthesized, and a higher score suggesting greater complexity or difficulty. We report both histograms of SA and a summary statistic in terms of the Wasserstein distance between the generated and reference distributions (smaller better).

13. **Quantitative estimation of drug-likeness (QED)** A metric designed to provide a quantitative measure of how 'drug-like' a molecule is. It essentially refers to the likelihood that a molecule possesses properties consistent with most known drugs, estimated based on a variety of molecular descriptors. We report both histograms of QED and a summary statistic in terms of the Wasserstein distance between the generated and reference distributions (smaller better).

14. **Molecular weight (Weight)** The sum of atomic weights in a molecule. We report both histograms of molecular weights and a summary statistic in terms of the Wasserstein distance between the generated and reference distributions (smaller better).

