# OpenReview forum: "Heterophily-informed Message Passing"
_TMLR — Accepted by TMLR_

### Review · Reviewer_6iA2 · 2025-01-28

**Summary Of Contributions:**

In this work, the authors propose two new message-passing (MP) approaches, one designed for homophilic and one designed for heterophilic tasks, augmenting the standard message-passing on the original graphs that takes all neighbors of a node into account. Importantly, these MP approaches do not require access to the label distribution of the underlying tasks, as homophily and heterophily are incorporated based on the similarity of node representation instead. As a result, the MP approaches are applicable to applications where no label information is available, e.g., in the case of private or missing labels or in generative modeling applications. The MP approaches are evaluated on homophilic and heterophilic node classification tasks, as well as on molecular generation benchmarks.

**Audience:**

Yes

**Claims And Evidence:**

No

**Requested Changes:**

* A concrete and detailed explanation of (a) why existing (heterophily-aware) do not effectively mitigate oversmoothing and (b) how these issues are overcome by the proposed approach. In addition, a concrete and detailed explanation how existing heterophily-aware GNNs lack the flexibility and applicability of the proposed approach.  *This is critical to my recommendation*.

* A more nuanced discussion of the empirical results on node classification that reflects the fact that only rarely do the proposed homophilic and heterophilic MPs show clear benefits on homophilic and heterophilic tasks, respectively. *This is critical to my recommendation*.

* I think that an empirical analysis on whether oversmoothing occurs in the base GNNs and whether it is effectively mitigated by the proposed homophilic and heterophilic MPs would strengthen the work but is not critical to my recommendation.

In case the authors feel that I have misunderstood parts of their work, I am happy to discuss the weaknesses above.

**Strengths And Weaknesses:**

## Strengths
* I appreciated the motivation of the authors to enable their heterophily-aware GNN to applications beyond node classification, e.g., for the proposed molecular generation application.
* Connected to the first point, the fact that the authors evaluate both in a predictive and generative setting increases the potential interest in this work and allows for a more rigorous evaluation of the proposed method.
* The node classification setting is rigorously evaluated, using a high number of random seeds and data splits, as well as a fair model comparison.

## Weaknesses
### **W1**: The work is poorly motivated.

The main contribution of this work is a novel variant for message-passing to mitigate oversmoothing. However, oversmoothing is barely introduced and more importantly, existing approaches to mitigate oversmoothing are not discussed in detail, neither are reasons why standard message-passing leads to oversmoothing. As a result, after reading introduction, related work and Section 3.1. and 3.2., it does not become clear what concrete issues persist in existing approaches that warrant the novel message-passing approach proposed by the authors. For example, at the end of 3.1., the authors write

> However, in practice, a naïve aggregation strategy typically mixes messages, especially in heterophilic areas, leading to the ‘oversmoothing’ problem

followed by citations of various previous works. Here, the authors do not disucss *why* simply mixing homophilic and heterophilic messages leads to oversmoothing. As a result, it becomes difficult to understand why the new message-passing approach, presented in Section 3.2., overcomes these limitations and prevents oversmoothing.

Another example is the following excerpt from Section 2, when discussing other heterophily-aware GNNs:

> These methods typically require additional parameters or specialized structures to achieve their effects. However, we hope to capture the effects with minimal modifications (no additional parameters), making the solution flexible and widely applicable.

It is not clear to me why these additional parameters hinder the application of the mentioned methods to e.g., molecular generation, such as the approach presented by the authors, nor why they make the methods less flexible.

### **W2**: Lack of clear empirical evidence for effectiveness of proposed approach
The results in Table 1 do not convince me that the proposed method offers real benefits over the base GNN models. For example, for the GCN, GCN+het. wins on Cornell but is on par or looses to the base GCN on all other heterophilic datasets, despite the fact that we know that these datasets are heterophilic. In addition, the GCN+het. looses to GCN+hom. on Chameleon, meaning that the homophilic variant wins over the heterophilic variant on a heterphilic dataset. The inverse case also exists, where GraphSAGE+het. beats GraphSAGE+hom. on CoAuthorCS, and GIN+het. beats GIN+hom. on Computers, CoAuthorCS and Photo. As such, it appears to me that there is no *clear* signal that the heterophilic MP improves performance on heterophilic datasets and that the homophilic MP improves performance on the homophilic datasets.

This also somewhat weakens the claim made by the authors that

> for the 6 heterophilic data sets, our MP schemes perform as well or better then the baseline (in 24 out of 24 cases)

This statement is true, yet only if one considers both heterophilic and homophilic MP jointly. Just looking at the performance of the heterophilic MP compared to the two other models on these datasets, the heterophilic MP is only on-par or better on 17/24 cases and beats both other models only in a *single* case: Cornell with GCN.

To summarize, my concern with the benchmark results is less with the performance of the homophilic MP and heterophilic MP per se, but with the fact that the heterophilic or homophilic nature of a particular dataset does not seem to be related all too well to whether the heterophilic or homophilic MP offers improvements.

### **W3**: Oversmoothing is never actually evaluated
While mitigating oversmoothing is a central motivation of this work, it is never actually measured on any of the experiments. Is oversmoothing even a concern on the evaluated node classification datasets? And if so, do the homophilic or heterophilic MPs actually mitigate it? Such an analysis could perhaps also shed light on the results mentioned in **W2**, where the homophilic MP outperforms heterophilic MP on heterophilic datasets and vice versa.

---

> ### Author Response · Authors · 2025-03-03
>
> We appreciate your valuable feedback, which has helped clarify the message in the paper. We address the comments and concerns below.
>
> > Oversmoothing is barely introduced and reasons why standard message-passing leads to oversmoothing
>
> Oversmoothing is a common phenomenon that appears in graph neural network domains, it means the node embeddings get less distinguishable after several GNN layers, which makes the representation less meaningful, and adversely affects the performance of downstream tasks.  The message passing scheme is similar to a heat conduction process on a special manifold, which maximizes the space entropy after time. For example, the classical GCN [1] can be seen as a form of Laplacian smoothing, which leads to oversmoothing after several layers by nature. We provide an example visualization of it in the 'original channel' in Fig. 1. This phenomenon has been well-analyzed in literature (such as [2]) and we try to keep the presentation concise.
>
> > Existing approaches to mitigate oversmoothing are not discussed in detail.
>
> The graph heterophily and oversmoothing problem is well-known in literature (e.g., [3, 4]). The nodes in a heterophilous graph tend to have neighbours more different from themselves and lose more distinguishability due to oversmoothing, leading to worse representations. It means we can mitigate the oversmoothing issue by incorporating more heterophilous prior into the model structure. Our method is inspired by heterophily, thus we discuss the heterophily-aware methods in this paper.
>
> > Here, the authors do not disucss why simply mixing homophilic and heterophilic messages leads to oversmoothing.
>
> There might be a misunderstanding on 'homophilic messages' or 'heterophilic messages' here. Homophily and heterophily measure the graph property, it measures the preference of nodes in such graphs to connect to similar nodes. The message is related to the GNN or "message passing neural networks", which collect the 'message' from all source nodes as shown in Eq. (1), and propagate them to all destination nodes (typically there is an order-invariant function to process all messages from the neighbours of the destination nodes) as shown in Eq. (2). There is no mixing of homophilic and heterophilic messages. The oversmoothing problem comes from the message-passing structure as introduced before.
>
> > As a result, after reading introduction, related work and Section 3.1. and 3.2., it does not become clear what concrete issues persist in existing approaches that warrant the novel message-passing approach proposed by the authors
>
> Fig. 1 gives one simple example of why our proposed scheme can mitigate the embeddings becoming less distinguishable by comparing three channels. The original channel is the basic message passing, which averages all features faster than the other two novel channels.
>
> > However, in practice, a naïve aggregation strategy typically mixes messages, especially in heterophilic areas, leading to the ‘oversmoothing’ problem
>
> It means the traditional message passing will mix all messages from all neighbour nodes, ignoring the local homophily on different nodes. The oversmoothing could be mitigated if we could account for such information.
>
> >  As a result, it becomes difficult to understand why the new message-passing approach, presented in Section 3.2., overcomes these limitations and prevents oversmoothing.
>
> Fig. 1 shows that, in the same depth of the homophilous and heterophilous channels, the node embeddings are much more distinguishable than the original channel, which is the same as the traditional GNN MP.
>
> > It is not clear to me why these additional parameters hinder the application of the mentioned methods to e.g., molecular generation, such as the approach presented by the authors, nor why they make the methods less flexible.
>
> Yes, we agree that this could be clarified. Thus, we now rephrase this in the paper as: "These methods typically ignore the local homophily during message passing or require specialized structures to achieve their effects". In addition, now we introduce the mixed channel approach, which requires more parameters.
>
> **References:**
>
> [1] Kipf, Thomas N., and Max Welling (2016). "Semi-supervised classification with graph convolutional networks." arXiv preprint arXiv:1609.02907.
>
> [2] Li, Qimai, Zhichao Han, and Xiao-Ming Wu. (2018). "Deeper insights into graph convolutional networks for semi-supervised learning." Proceedings of the AAAI conference on artificial intelligence. Vol. 32. No. 1.
>
> [3] Bodnar, Cristian, et al. (2022) "Neural sheaf diffusion: A topological perspective on heterophily and oversmoothing in gnns." Advances in Neural Information Processing Systems (NIPS) 35: 18527-18541.
>
> [4] Yan, Yujun, et al. (2022). "Two sides of the same coin: Heterophily and oversmoothing in graph convolutional neural networks." 2022 IEEE International Conference on Data Mining (ICDM). IEEE.

---

> ### Author Response · Authors · 2025-03-03
>
> > Lack of clear empirical evidence for effectiveness of proposed approach
>
> We fixed some minor things in the code, added new data sets, added new homophily metrics (as suggested by Fdni), and introduced the mixed results (as suggested by ALAZ). In the manuscript, the node experiments and discussion sections are updated. The updated experiment results in Table 1 and Fig. 2 demonstrate that the MixMP enhances the node classification performance with a 3.84 %-point accuracy improvement on average over all runs.
>
> > Oversmoothing is never actually evaluated
>
> The main concern of oversmoothing is the less distinguishable representations could influence the downstream task like node classification and graph generation. Fig. 1 gives a visual example of how the oversmoothing phenomenon is mitigated by heterophilous and homophilous channels rather than the original channel, which motivates us to design our structure. Moreover, oversmoothing is a phenomenon which serves as the intuition for model improvement, which is hard to evaluate quantitatively. The performance improvement, especially on heterophilious data, could empirically show the nodes distinguishability is improved by our method.

---

### Review · Reviewer_ALAZ · 2025-01-31

**Summary Of Contributions:**

In this paper, the authors propose a novel message-passing scheme for heterophilic graphs. Given any base MP scheme, the authors produce two new "modes", homophilic or heterophilic, simply by scaling all messages by the cosine similarity (or one minus the cosine similarity) of the representations of the nodes in the previous layer. They test these different "modes" on four classical GNN architectures, and on a normalizing flow architecture for molecular generation, where they report improved result.

**Audience:**

Yes

**Claims And Evidence:**

Yes

**Requested Changes:**

See above: more general architecture, experiments against other heterophilic GNNs

**Strengths And Weaknesses:**

Strengths:
- a very simple scheme that can be adapted to any MP architecture
- experiments on graph generation, which is quite original in this literature

Weaknesses:
- although I like the simplicity of the idea, I think it could be pushed a lot further. For instance, it seems the authors introduces three distinct modes, which are tested completely separately? (Unless I am mistaken?) Why not propagate all three channels, then let the GNN learn how to combine them? This would mitigate the need to know "in advance" if the dataset is heterophilic of homophilic.
- In the same fashion, there are probably other type of "similarity" notions that could be tested for the structured MP
- many "heterophilic" strategies are mentioned in the intro, but are not tested in the experiments, where the proposed scheme are only tested against the vanilla GNN versions. It could be useful to test a few of them.

---

> ### Author Response · Authors · 2025-03-03
>
> We thank reviewer ALAZ for the insightful feedback, which aided us in clarifying aspects of this work. We address the comments and concerns below.
>
> > For instance, it seems the authors introduces three distinct modes, which are tested completely separately? (Unless I am mistaken?) Why not propagate all three channels, then let the GNN learn how to combine them?
>
> Yes, they are tested separately. Thank you for your suggestion; we have now introduced a mixed version, which propagates and combines the three channels. So that we don't need to know in advance if the data set is heterophilic or homophilic. The updated experiment results in Table 1 and Fig. 2 demonstrate that the MixMP enhances the node classification performance with a 3.84 %-point accuracy improvement on average over all runs.
>
> > In the same fashion, there are probably other type of "similarity" notions that could be tested for the structured MP.
>
> We agree that there are many potential designs of similarity. We chose the cosine similarity because of the computational efficiency and no requirement for additional knowledge.
>
> > Many "heterophilic" strategies are mentioned in the intro, but are not tested in the experiments, where the proposed scheme are only tested against the vanilla GNN versions. It could be useful to test a few of them.
>
> Currently, advanced GNN-based models (e.g., in graph generation) only utilize vanilla GNNs as model blocks for structure simplicity. We aim to design a simple module which is easy to plug in. Thus we compare our proposed scheme against the most used GNNs, such as GCN and GraphSAGE. Most 'heterophilic' strategies change the message passing pipelines much different from traditional GNNs, which are hard to merge with our scheme. To present a comparison with other heterophily-aware baselines, we select GraphSAGE+mix as one example to compare with other heterophilic models in App. B4. We achieve better results than the base model GraphSAGE but do not beat the state-of-the-art ACM-GCN method [1].
>
> [1] Luan, S., Hua, C., Lu, Q., Zhu, J., Zhao, M., Zhang, S., ... & Precup, D. (2022). Revisiting heterophily for graph neural networks. Advances in Neural Information Processing Systems (NeurIPS), 35, 1362-1375.

---

### Review · Reviewer_Fdni · 2025-02-17

**Summary Of Contributions:**

The paper presents a new method for considering homophily and heterophily in message passing. In particular, the method proposes scaling the messages of each GNN layer by a function that depends on a global term gamma (corresponding to homophily or heterophily of the graph), and an edge-dependent term H(u,v) which computes the nodes' cosine similarity.

This method is applied to 4 GNN models (GCN, GAT, GIN, GraphSAGE) on a range of node classification benchmarks that vary in how homophilic they are based on their node homophily metric. In each instance, the original GNN is compared to its homophilic version (where all messages are scaled by the edge similarity) and its heterophilic version (where all messages are scaled by 1-edge similarity). In most combinations (GNN x dataset), either the homophilic or heterophilic version matched or outperformed the original GNN, with the homophilic version being more advantageous on homophilic datasets and similarly for the heterophilic; however, GAT and GIN provide exceptions to this intuition.

The method is also applied in combination with MoFlow, a normalizing flow model for molecular graphs. This model has an atom flow and a bond flow,  and it modifies the message passing in the same way as for the node classification GNNs. The model is tested on QM9 and ZINC, and compared with the original MoFlow version and GraphDF, matching their performance or exceeding, especially when sampling the adjacency matrix from the real data distribution, instead of using the bond flow.

**Audience:**

No

**Broader Impact Concerns:**

I do not have any concerns to raise on the ethical implications of the work.

**Claims And Evidence:**

Yes

**Requested Changes:**

Adjustments that are critical to securing my recommendation for acceptance:

I think the proposed method needs a more thorough integration of the findings from previous works addressing heterophily for GNNs. In particular, the method should be compared with other methods designed for heterophily (e.g. [2] or other related work). Similarly, previous papers have found limitations with some of the used datasets for evaluating heterophily (the [1] mentioned above) -- evaluating on the proposed benchmark could be a better solution. Lastly, both [1] and [2] mention the drawbacks of node homophily, and consider other metrics (e.g. adjusted homophily). As it is unclear how node homophily relates to the homophilic and the heterophilic versions of the GNNs proposed, it is possible that other metrics would provide a clearer picture on which module the community should use depending on how the dataset is characterized.

Adjustments that, in my view, would strengthen the work:

There are three types of clarifications that could improve the readability of the paper:
- Gamma: Gamma is a factor that is used for scaling each message, at each layer, but, if I understand correctly, it is treated as a hyper-parameter. However, the ablation study mentions that "the three channels could share parameters", from which I understand that all three versions (original, homophilic and heterophilic) are used together. It would be useful to clarify when gamma is set (as well as what other alternatives could be), and if different values of gamma are combined, how is the combination computed.
- MoFlow: it would be useful to mention why MoFlow was chosen and if the validity correction used in the original MoFlow paper was used here as well.
- 'true adjacency': it would be easier to follow if this was introduced before the results, and also if it was explained how the bond flow was used in this case.

[2] Luan, Sitao, et al. "Revisiting heterophily for graph neural networks." Advances in neural information processing systems 35 (2022): 1362-1375.

**Strengths And Weaknesses:**

The paper's main strength is that it proposes a general modification to the GNN, that can customise it either for a homophilic or a heterophilic use case, without introducing additional parameters or significant additional computation. The writing is generally clear, and the method is tested on different GNN x dataset combinations for node classification, as well as for graph generation.

The main weakness is that it is not clear how the findings can be used by the community -- the heterophilic datasets used are known to have some limitations (see [1], where also new datasets were proposed), the method is not compared with other approaches for tackling heterophily for GNNs, and the results don't have a clear suggestion on which the version of the module should be used (heterophilic, homophilic, original) in relation to the homophily of the dataset. Additional weaknesses are related to clarity and providing additional explanations, which I will describe in the next section.

[1]: Platonov, Oleg, et al. "A critical look at the evaluation of GNNs under heterophily: Are we really making progress?." arXiv preprint arXiv:2302.11640 (2023).

---

> ### Author Response · Authors · 2025-03-03
>
> We thank reviewer Fdni for the insightful feedback, which helped in clarifying crucial aspects of this work. We address the comments and concerns below.
>
> > The method should be compared with other methods designed for heterophily
>
> We have aimed to design a simple module which is easy to plug into different models. Thus we compare our proposed scheme against the most used GNNs, such as GCN and GraphSAGE. To present a comparison with other heterophily-aware baselines, we selected GraphSAGE+mix as an example to compare with other heterophilic models. This additional experiment is now included in App. B4 in the revised manuscript. We achieve better results than the base model GraphSAGE but fail to beat the SOTA method ACM-GCN method [3] without a dedicated design on a combination of our scheme with the SOTA baselines.
>
> > Similarly, previous papers have found limitations with some of the used datasets for evaluating heterophily (the [1] mentioned above) – evaluating on the proposed benchmark could be a better solution
>
> We have added the data sets from [2] as an additional benchmark as a comparison (see Table 1). our mixMP model is stably better on datasets Minesweeper Roman-empire, and comparable on Questions, Amazon-ratings and Tolokers
>
> > It is possible that other metrics would provide a clearer picture of which module the community should use depending on how the dataset is characterized.
>
> We agree that this can be insightful. Following this suggestion, we have added the class insensitive edge homophily ratio $\mathcal{H}_{ei}$ [1] as the new homophily measure (see Table 1). The different homophily statistics are also visible in Table A4 in the Appendix in the revised manuscript. The visual patterns in Fig. 2 are changed to be clearer.
>
> >  It would be useful to clarify when gamma is set (as well as what other alternatives could be), and if different values of gamma are combined, how is the combination computed.
>
> The gamma is the hyperparameter of the message passing in the node classification task. In the ablation study of molecular generation, the HetFlows contains GNNs with three channels mixed by a linear layer with details in App. A5. We can share the parameters of message passing of the three channels and only change the hyperparameters in each channel. The parameter sharing is described in paragraph 'Ablation study: parameter sharing of heterophilous GNN' in Sec. 4.3. App. A2 and A5 provide the details of the HetFlows and the mixMP version GCN contained by HetFlows. Now we introduce the mixed model in the first paragraph in Sec. 3.2 and describe them in detail in App. A5.
>
> > MoFlow: it would be useful to mention why MoFlow was chosen and  if the validity correction used in the original MoFlow paper was used here as well.
>
> We selected MoFlow because it is a normalizing flow-based generative model. Compared to other generative models, it generates graphs that provide uncertainty estimation, which are compatible with the application of Bayesian tools. Validity correction is not used in our experiments, both for the baseline and our HetFlows. We mentioned it in the first paragraph in Sec. 3.3 "Application to Molecular Generation".
>
> > 'true adjacency': it would be easier to follow if this was introduced before the results, and also if it was explained how the bond flow was used in this case.
>
> The bond flow is used for generating the molecular graph structure. When the graph topology is sampled from the true adjacency distribution, the bond flow is not used in this case. We mentioned it in Sec. 3.3 in the revised manuscript.
>
>
> **References:**
>
> [1] Lim, D., Hohne, F., Li, X., Huang, S. L., Gupta, V., Bhalerao, O., & Lim, S. N. (2021). Large scale learning on non-homophilous graphs: New benchmarks and strong simple methods. Advances in neural information processing systems (NeurIPS), 34, 20887-20902.
>
> [2] Platonov, O., Kuznedelev, D., Diskin, M., Babenko, A., & Prokhorenkova, L. (2023). A critical look at the evaluation of GNNs under heterophily: Are we really making progress?. arXiv preprint arXiv:2302.11640.
>
> [3] Luan, S., Hua, C., Lu, Q., Zhu, J., Zhao, M., Zhang, S., ... & Precup, D. (2022). Revisiting heterophily for graph neural networks. Advances in Neural Information Processing Systems (NeurIPS), 35, 1362-1375.

---

### Author Response · Authors · 2025-03-03
**Reply to all reviewers**

We thank the reviewers for going through our manuscript and providing detailed comments. We appreciate the views and provided suggestion for improving the work.

The reviewers appreciated the clarity and generality of the proposed approach. We also see this as a key strength and rather than trying to focus on beating the current SOTA in particular use cases, this paper presents a general and easy-to-grasp building block.

We summarize the main changes at this stage:

- We have updated the node classification experiment (Sec. 4.1) as suggested by several reviewers. We introduce 5 new heterophilous graph data sets (roman-empire, amazon-ratings, minesweeper, tolokers, and questions) from [1] as suggested by reviewer Fdni, replace the node homophily with the class insensitive edge homophily ratio from [2] as suggested by reviewer Fdni, and add the mixMP mode for comparison as suggested by reviewer ALAZ. We re-ran the experiments with the modifications and changed some hyperparameters such as the drop-out ratio from 0 to 0.2, and GNN layers from 4 to 2. Table 1 and Figure 2 are updated with the new experiment results.
    - The mixMP is the combination of three channels (homMP, hetMP, and the original channel) with an additional linear layer such that we don't need to know the data homophily in advance. Then we introduce additional parameters and lose these benefits so the corresponding contents are revised.
    - The updated experiment results in Table 1 and Fig. 2 demonstrate that the MixMP enhances the node classification performance with a 3.84 %-point accuracy improvement on average over all runs.
- To present a comparison with other heterophily-aware baselines, we selected GraphSAGE+mix as an example to compare with other heterophilic models. This additional experiment is now included in App. B4 in the revised manuscript.
    - We achieve better results than the base model GraphSAGE but do not beat the SOTA method ACM-GCN method [3] without a dedicated design on a combination of our scheme with the SOTA baselines.
- We have clarified many parts of the paper, in particular:
    - Explained the graph generation settings `+true adj.` and heterophily-informed MP hyperparameter $\gamma$.
    - Rephrased the motivation to propose our scheme as "capture the local homophily difference with minimal structure modification, making the solution flexible and widely applicable."
- Additionally, we have carefully gone through the submission and made other minor adjustments to improve clarity and made typo corrections.

We provide detailed responses to the individual comments and concerns under each review separately. Additionally, we provide a revision of our manuscript that shows a difference from the previous version (additions in blue).


**References:**

[1] Platonov, O., Kuznedelev, D., Diskin, M., Babenko, A., & Prokhorenkova, L. (2023). A critical look at the evaluation of GNNs under heterophily: Are we really making progress?. arXiv preprint arXiv:2302.11640.

[2] Lim, D., Hohne, F., Li, X., Huang, S. L., Gupta, V., Bhalerao, O., & Lim, S. N. (2021). Large scale learning on non-homophilous graphs: New benchmarks and strong simple methods. Advances in neural information processing systems (NIPS), 34, 20887-20902.

[3] Luan, S., Hua, C., Lu, Q., Zhu, J., Zhao, M., Zhang, S., ... & Precup, D. (2022). Revisiting heterophily for graph neural networks. Advances in Neural Information Processing Systems (NeurIPS), 35, 1362-1375.

---

### Decision · Action_Editor_WdTq · 2025-03-27

**Recommendation:** Accept as is

**Comment:**

After a successful rebuttal, the reviewers unanimously agree that the paper has sufficiently progressed and is in a good shape for acceptance into TMLR now. I concur with the reviewers -- well done!

**Audience:**

Yes, this paper would clearly interest the sizeable graph representation learning audience within TMLR.

**Claims And Evidence:**

Yes, after the rebuttal revision, the paper clearly meets the bar of TMLR's technical soundness.